# ML-Tool-Bench: Tool-Augmented Planning for ML Tasks

## Abstract

The development of autonomous machine learning (ML) agents capable of end-to-end data science workflows represents a significant frontier in artificial intelligence. These agents must orchestrate complex sequences of data analysis, feature engineering, model selection, and hyperparameter optimization, tasks that require sophisticated planning and iteration. While recent work on building ML agents has explored using large language models (LLMs) for direct code generation, tool-augmented approaches offer greater modularity and reliability. However, existing tool-use benchmarks focus primarily on task-specific tool selection or argument extraction for tool invocation, failing to evaluate the sophisticated planning capabilities required for ML Agents. In this work, we introduce a comprehensive benchmark for evaluating tool-augmented ML agents using a curated set of 61 specialized tools and 15 tabular ML challenges from Kaggle. Our benchmark goes beyond traditional tool-use evaluation by incorporating an in-memory named object management, allowing agents to flexibly name, save, and retrieve intermediate results throughout the workflows. We demonstrate that standard ReAct-style approaches struggle to generate valid tool sequences for complex ML pipelines, and that tree search methods with LLM-based evaluation underperform due to inconsistent state scoring. To address these limitations, we propose two simple approaches: 1) using shaped deterministic rewards with structured textual feedback, and 2) decomposing the original problem into a sequence of sub-tasks, which significantly improves trajectory validity and task performance. Using GPT-4o, our approach improves over ReAct by 16.52 percentile positions, taking the median across all Kaggle challenges. We believe our work provides a foundation for developing more capable tool-augmented planning ML agents.

## 1 Introduction

Autonomous agents capable of solving end-to-end machine learning (ML) tasks represent a critical frontier in artificial intelligence (Grosnit et al., 2024; Toledo et al., 2025; Yang et al., 2025; Chan et al., 2025). Such agents must be capable of doing: data preprocessing, feature engineering, model training, and hyperparameter tuning, while managing intermediate results and adapting their strategies based on the evolving context. Achieving this level of autonomy requires not only sophisticated planning, but also memory management and the capacity to coordinate multiple operations coherently. Large language models (LLMs) have recently been explored as the foundation for such agents (Grosnit et al., 2024; Chan et al., 2025; Huang et al., 2024). Early work has primarily focused on direct code generation, where the agent generates python code for completing a given ML task (Grosnit et al., 2024; Chan et al., 2025; Huang et al., 2024; Toledo et al., 2025). This paradigm has shown promise on competitive benchmarks inspired by Kaggle challenges, with some approaches achieving performance comparable to a Kaggle Master (Grosnit et al., 2024; Chan et al., 2025). Several benchmarks have also been proposed to evaluate the performance of LLMs on such tasks (Chan et al., 2025; Huang et al., 2024; Qiang et al., 2025; Jing et al., 2025; Zhang et al., 2025). However, any approach that relies on direct code generation is prone to key weaknesses: generated code is brittle (Abbassi et al., 2025; Liu et al., 2025), debugging typically requires multiple iterations, and reasoning is tightly coupled with execution (Liu et al., 2025; Chen et al., 2025).

An alternative paradigm equips LLMs with external tools, yielding tool-augmented agents that need to decide which tools to invoke and in what sequence, to solve the task. Tools offer modular, reusable

building blocks for data-science workflows: from preprocessing, to training, and evaluation. This design has proven effective in broader domains, including web navigation (Zhou et al., 2024b), operating systems (Bonatti et al., 2024), and code interpretation (Huang et al., 2024), yet its potential for ML workflows remains underexplored. Crucially, tool augmentation reformulates the problem as planning in a large action space: the agent must coordinate multi-step trajectories and retrieve and reuse intermediate artifacts (or results). Because the agent is restricted to a curated toolset, tool-augmented approaches decouple high-level reasoning from low-level code execution, improving modularity, reliability, and safety.

Existing benchmarks for tool use fall short on long-horizon planning. Most benchmarks and approaches evaluate whether agents can select the right tools and valid arguments. The Berkeley Function-Calling Leaderboard (BFCL) (Patil et al., 2025) measures single, parallel, and multiple function calling, and BFCL-v3 (Patil et al., 2025) extends this to multi-turn, multi-step settings. However, even BFCL-v3 emphasizes relatively shallow plans compared to ML workflows, which might require long-term planning, iterative refinement, and reuse of intermediate artifacts. Similarly ToolBench (Xu et al., 2023) provides a suite of diverse software tools, that span both single-step and multi-step action generation, but focuses on evaluating whether the LLM can correctly select tools and tool arguments.

In this work, we introduce ML-Tool-Bench, motivated by the lack of good benchmarks to assess planning approaches with tools in ML workflows. In particular, ML-Tool-Bench provides a benchmark to evaluate the planning capabilities of LLM agents on *tabular* Kaggle ML challenges. We introduce a curated suite of 61 tools sufficient to solve such tasks and assess performance across 15 Kaggle challenges spanning regression and classification.

We evaluate multiple agents using several different planning algorithms, on our benchmark. To enable agents to create, persist, and reuse intermediate artifacts, we adopt an in-memory, named-object management scheme: tools accept references to named objects, and agents can assign names to tool outputs. We refer to this as scratchpad-augmented planning: agents store and retrieve objects by name over multi-step trajectories, enabling tools to handle arbitrarily large or structured inputs, unlike prior benchmarks that restrict arguments to simple types (e.g., strings, integers, floats). We observe that simple methods like ReAct (Yao et al., 2023b) struggle to produce performant trajectories across our Kaggle benchmark. Monte Carlo Tree Search-based methods (Kocsis & Szepesvari, 2006; Silver et al., 2016) such as LATS (Zhou et al., 2024a), which rely on LLMs as value estimators, also underperform due to inconsistent trajectory scoring. In contrast, we propose two simple approaches: 1) combining shaped, deterministic rewards with textual feedback and 2) decomposing the original problem into a sequence of sub-tasks. These approaches outperform the baselines, yielding more performant tool trajectories. These results highlight the difficulty of autonomous ML planning and point toward tool-augmented systems that rely less on subjective LLM scoring as tool sets grow in size and complexity.

1. We introduce ML-Tool-Bench, a tool-augmented benchmark for end-to-end ML planning with 61 tools and 15 Kaggle challenges.
2. We formalize *scratchpad-augmented planning* via named-object management that supports arbitrarily large artifacts and reversible branching in search.
3. We propose *MCTS-Shaped*, an MCTS approach with shaped, deterministic rewards and targeted textual feedback, which improves trajectory validity and performance over ReAct and LATS.
4. We introduce *Hierarchical MCTS*, an approach that decomposes problems into sequenced sub-tasks, further improving validity and robustness. For GPT-4o, Hierarchical MCTS improves over LATS by 9.93 percentile positions on the leaderboard and over ReAct by 16.52 percentile positions (median across all competitions). For GPT-4.1-mini, it improves over MCTS-Shaped by 1.89 percentile positions, while both ReAct and LATS had a median percentile position of 0.

Together, these advances establish strong baselines for tool-augmented, end-to-end ML planning and reduce reliance on subjective LLM scoring.

## 2 RELATED WORK

**Machine Learning Benchmarks for AI Agents**: Most of the existing Data Science and ML benchmarks, provide the LLM agent access to write code that solves the task, and evaluate its performance. Chan et al. (2025) propose MLE-bench, a curated benchmark of 75 Kaggle challenges, that test real-

world ML engineering skills. They find that OpenAI's o1-preview with the AI-Driven Exploration (AIDE) scaffolding (Jiang et al., 2025) achieves at least a level of Kaggle bronze medal in 16.9% of competitions in their benchmark. AIRA-dojo (Toledo et al., 2025) improves upon Chan et al. (2025), replacing AIDE (Jiang et al., 2025) with a different choice of operator set, to generate new candidate solutions, and using Monte Carlo Tree Search (MCTS) (Kocsis & Szepesvari, 2006) instead of greedy search, increasing the success rate of achieving a Kaggle medal from 39.6% to 47.7%. Huang et al. (2024) also propose a ML benchmark, called MLAgentBench, containing a suite of 13 tasks, where the agent is allowed to perform actions like read/write files, execute code and inspect outputs. They construct a ReAct based agent (Yao et al., 2023b) (with Claude v3 Opus) and were able to build compelling ML models on MLAgentBench with 37.5% average success rate. Qiang et al. (2025) propose MLE-Dojo, an interactive gym-style workflow for LLM agents in iterative ML engineering workflows, and build upon 200+ Kaggle challenges. To evaluate Data Science Agents, Jing et al. (2025) proposed a comprehensive benchmark that includes 466 data analysis tasks and 74 data modeling tasks, sourced from Eloquence and Kaggle competitions, and showed that state of the art LLMs and agents struggle on most tasks. Zhang et al. (2025) propose DataSciBench and demonstrate that closed source models (GPT, Claude etc.) outperform open source models on all metrics in their benchmark.

**Learning in Tool augmented LLMs**: Solving ML challenges solely through the invocation of a fixed set of tools, in the correct sequential order, remains relatively unexplored. Approaches such as ARTIST (Singh et al., 2025), ReTooL (Feng et al., 2025), StepTool (Yu et al., 2024), ToRL (Li et al., 2025), and ToolPlanner (Wu et al., 2024) couple reasoning and tool use for LLMs, using Reinforcement Learning to learn robust strategies for tool use. Recently, methods to fine-tune LLMs on responses containing tool usage have also been proposed (Schick et al., 2023; Qin et al., 2023; Gou et al., 2023; Patil et al., 2023).

Alternately, tree search methods (Yao et al., 2023a; Hao et al., 2023; Zhou et al., 2024a; Zhuang et al., 2023) have also been used to generate valid tool use trajectories. Zhuang et al. (2023) employs A* search, Hao et al. (2023) adopts Monte Carlo Tree Search (MCTS) and uses LLM as the world model, Zhou et al. (2024a) uses MCTS with value functions obtained from an LLM and self-reflection, and Yao et al. (2023a) explores Breadth-First Search (BFS) and Depth-First Search (DFS). However, these methods either depend on heuristic cost functions or leverage LLM feedback as a value function, and they are primarily applied to problems with relatively shallow depth. LATS (Zhou et al., 2024a) and Toolchain* (Zhuang et al., 2023) are the only approaches that explore planning with tools while the others restrict themselves to reasoning or toy domains. Feng et al. (2024) propose TS-LLM, an AlphaZero-inspired tree-search framework for LLMs that integrates a learned value function to guide decoding. The trajectories generated from tree search can further be used to fine-tune and improve the LLM, and TS-LLM has been shown to scale to tree depths of up to 64. Another approach, ReST-MCTS (Zhang et al., 2024), adopts a similar strategy to TS-LLM; however, in this case the per-step rewards are inferred directly from MCTS, whereas TS-LLM infers them using TD-$\lambda$ (Sutton, 1988).

**Tool Benchmarks**: Benchmarks for LLM tool use largely emphasize correct tool selection and argument specification rather than extended planning. ToolBench (Xu et al., 2023) covers diverse software tools for single- and multi-step tasks but underplays long-horizon coordination. The Berkeley Function Calling Leaderboard (BFCL) (Patil et al., 2025) evaluates single, parallel, and multi-step calls, though plans remain shallow. $\tau$-Bench (Yao et al., 2024) focuses on human–agent interaction under domain rules, highlighting alignment and information gathering more than proactive planning.

## 3  ML-TOOL-BENCH

Each task in ML-Tool-Bench, can be formalized as a Markov Decision Process (MDP) $(\mathcal{S}, \mathcal{A}, \mathcal{T}, R)$ (Puterman, 2014) The *state space* $\mathcal{S}$ consists of the entire interaction history: all AI, Human, and Tool messages together with artifacts such as dataframes and ML models. Whenever a tool is executed, its observations (e.g., outputs, errors, logs) are appended to the history and folded into the state, so that the state maintains an up-to-date record of both conversational and artifact changes. The initial state $s_0$ comprises the Kaggle challenge description along with the dataset.

The *action space* is defined as: $\mathcal{A} = (\mathcal{A}_{\text{tool}} \cup \{\varnothing\}) \times (\mathcal{A}_{\text{reason}} \cup \{\varnothing\})$, where $\mathcal{A}_{\text{tool}}$ denotes the set of all tool invocations together with their full parameterizations (not just tool identity, but also

argument values and hyperparameters). This makes the benchmark challenging, since the effective size of $\mathcal{A}_{\text{tool}}$ can be very large rather than a small, discrete set. The set $\mathcal{A}_{\text{reason}}$ is the space of free-form reasoning steps, which we model as natural-language strings. The null element $\varnothing$ denotes "no action" in that component, allowing tool-only, reason-only, both, or neither at a step. Reasoning actions organize information, plan future steps, and inject prior knowledge; tool actions modify or analyze data and train/evaluate models, thereby updating the state's artifacts.

The *transition function* $\mathcal{T} : \mathcal{S} \times \mathcal{A} \rightarrow \mathcal{S}$ maps a state–action pair $(s, a)$ to the next state by appending the messages generated by the agent's action, appending tool messages (i.e., the observations produced), and updating artifacts accordingly.

The *reward function* $R$ evaluates progress and can be instantiated in several ways: (i) an outcome reward granted upon successful challenge completion; (ii) a shaped reward providing intermediate credit for measurable progress; or (iii) an LLM-based evaluation of the current state, using the LLM as a judge (Zheng et al., 2023) and absolving us from providing the reward function.

### 3.1 SCRATCHPAD

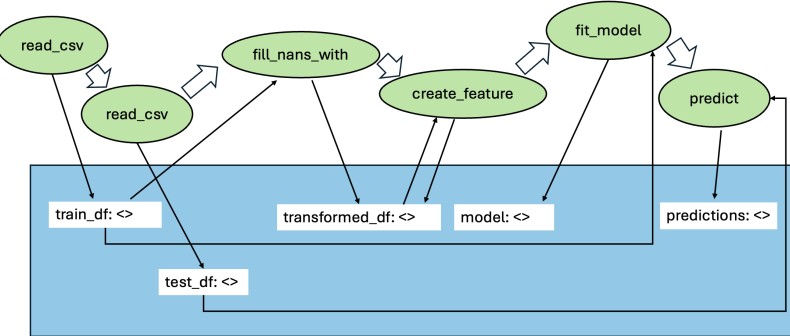

Figure 1: An illustration of our named-object management scheme. Green circles denote tool calls; the blue rectangle denotes the scratchpad (a key–value store). Each tool can read any named object from the scratchpad and write outputs back to it, depending on their read-write behavior. Arrows into a tool indicate inputs; arrows from a tool to the scratchpad indicate outputs. read_csv is a set tool; fill_nans_with, fit_model, and predict are get–set tools; create_feature is an override tool. There are two read_csv tool calls in the figure, one for train data and one for test.

Solving an ML challenge often involves storing large dataframes, models, and other complex artifacts as they cannot be directly passed as tool inputs by an LLM. A naive workaround is to maintain a single dataframe and model object that the agent incrementally modifies via tool calls. However, a single erroneous call can corrupt these objects, forcing a restart of the trajectory, and the agent becomes inflexible to create and reuse intermediate variables.

To address this, we adopt an in-memory, named-object management scheme: an agent assigns names to tool outputs, and tools accept references to named objects as inputs. Thus, agents can pass complex objects to tools by specifying the name under which the object is stored in the scratchpad. An illustration of this approach is presented in Figure 1. Implementing this requires modifying tools to operate on named references rather than raw objects; we describe these changes next.

### 3.2 TOOLS

We grant the agent access to a curated suite of 61 tools spanning data loading, data cleaning, feature engineering, and modeling. These tools are designed to be reasonably sufficient for solving tabular regression and classification tasks. Agent performance depends on the available toolset: in principle, a very large collection would maximize flexibility, but it results in an increased action space and complicates planning. We therefore adopt a fixed, compact tool set that trades some flexibility for a more tractable planning, while remaining adequate to solve the Kaggle challenges considered. For

modeling, we restrict to tree-based learners: Random Forest, XGBoost, LightGBM, and CatBoost, and linear/logistic regression, in light of the strong performance of tree-based methods on tabular Kaggle challenges (Grinsztajn et al., 2022). For more information on tools and how arbitrary user defined tools are modified to operate on named references rather than objects, refer to Appendix $E$

### 3.3 KAGGLE CHALLENGES

We select 15 tabular Kaggle ML challenges for ML-Tool-Bench: eight classification (binary and multiclass) and seven regression. These tasks are chosen so that they are solvable with our tool set. Several datasets are large (e.g., New York City Taxi Fare Prediction is ∼2.5 GB), so we randomly sample 10,000 data points from each competition's training set to keep planning computationally tractable. Because Kaggle test labels are hidden, we create an internal evaluation split by reserving 20% of the sampled training data as a test set with ground-truth labels. We evaluate using each competition's official metric and report agent performance as the corresponding public-leaderboard percentile. Our evaluation metric is chosen to accommodate a collection of regression and multi-class classification tasks. Note that Kaggle leaderboards are computed on a test set, the labels to which we do not have access to; our reported results are computed on our held-out test split. For more information on the Kaggle challenges, refer to Appendix $C$

## 4 APPROACHES

### 4.1 REACT

ReAct (Yao et al., 2023b) is a prompting framework that interleaves natural-language reasoning (*Thought*) with tool interaction (*Action*) and the subsequent *Observation* from the environment due to tool calling. ReAct augments the agent's action space to include the space of language, to account for thoughts or reasoning traces that do not affect the environment. Thoughts compose useful information from the current context and update the context to support future reasoning or actions. By explicitly exposing intermediate chain-of-thought alongside tool calls, ReAct enables agents to plan, invoke tools, and revise plans based on feedback. However, ReAct is unidirectional and can neglect potential alternative continuations from certain states, leading to locally optimal solutions (Zhuang et al., 2023; Zhou et al., 2024a).

### 4.2 MONTE CARLO TREE SEARCH (MCTS)

MCTS (Kocsis & Szepesvari, 2006) is a search algorithm that has achieved remarkable success in challenging domains such as Go (Silver et al., 2016) and Atari (Ye et al., 2021). MCTS builds a search tree where nodes correspond to states and edges correspond to actions. It comprises four phases: *selection*, *expansion*, *simulation/rollout*, and *backpropagation*. A common *selection* policy uses UCT (Upper Confidence Bound for Trees) (Kocsis & Szepesvari, 2006), choosing a child $s$ of parent $p$ such that: $s \in \arg\max_{s \in \mathbb{C}(p)} V(s) + w\sqrt{\ln N(p)/N(s)}$,

where $V(s)$ is the empirical value function, denoting the expected cumulative reward from state $s$, $N(p)$ is the parent's visit count, $N(s)$ is the child's visit count, $w > 0$ controls exploration, and $\mathbb{C}(p)$ denotes the set of children of $p$. Upon reaching a leaf node, it is *expanded* by selecting an action and adding the resulting next state as a child. From the newly expanded node, a *simulation* is run until the end of the episode or a fixed depth to obtain a reward $r$, which is then *backpropagated* along the trajectory to update values of all states along that trajectory: $V(s) \leftarrow (V(s)(N(s) - 1) + r)/N(s)$. MCTS is well-suited to large, irregular action spaces and provides a principled trade-off between exploration and exploitation. A pictorial illustration of MCTS is provided in Appendix $B$.

### 4.3 LANGUAGE AGENT TREE SEARCH (LATS)

LATS (Zhou et al., 2024a) adapts MCTS to language agents by using LLMs both to propose actions (reasoning steps or tool calls) and to evaluate node values. At each expansion, the policy LLM suggests candidates, and an evaluator LLM scores partial trajectories based on estimated progress toward the task objective. The value of a state is taken to be a weighted average of the evaluator LLM's score and a self-consistency score (Wang et al., 2022), which upweights frequent candidates

in the expansion stage. In our tool-planning setting, we do not incorporate the self-consistency score into the value of a state. We observed that during the expansion phase, the LLM tends to propose only a small but distinct set of tool calls or reasoning steps, making the additional score unnecessary. LATS has shown improvements over purely reactive methods, such as ReAct (Yao et al., 2023b) on complex tasks. However, its value estimates can be noisy, and the effective planning depth may be limited by inconsistencies in evaluator scoring.

### 4.4 MCTS-Shaped

In MCTS with shaped rewards, the agent receives intermediate credit for completing stages of the Kaggle ML challenge. The shaped-reward stages and their triggers are detailed below. Figure 2 provides an example to illustrate how rewards are provided in MCTS-Shaped.

**Shaped-reward stages**

1. **Train data loading**: reward when the agent successfully loads the training data.
2. **Test data loading**: reward when the agent successfully loads the test data. Note that test data does not have the target variable, that needs to be predicted.
3. **Combine train and test**: reward when the agent correctly concatenates train and test to enable consistent cleaning and feature engineering.
4. **Data cleaning**: reward when no missing values (NaNs) remain in the combined data.
5. **Feature engineering**: reward when (a) all categorical variables are properly encoded (e.g., one-hot or label encoding), and (b) the resulting feature dimensionality remains within a reasonable bound (to avoid exploding features from, e.g., high-cardinality text-like columns).
6. **Split back to train/test**: reward when the agent correctly splits the combined data back into train and test after transformations.
7. **Train features/target**: reward when the agent extracts $(X_{\text{train}}, y_{\text{train}})$ from the training dataframe using the correct target column.
8. **Test features**: reward when the agent extracts $X_{\text{test}}$ from the test dataframe (which prior to this stage contains a dummy target), with correct arguments.
9. **Modeling**: reward when the agent successfully fits a model on the training data; the reward is proportional to cross-validation performance.
10. **Create submission**: reward when the agent generates predictions on the test data and writes a valid submission CSV to disk.

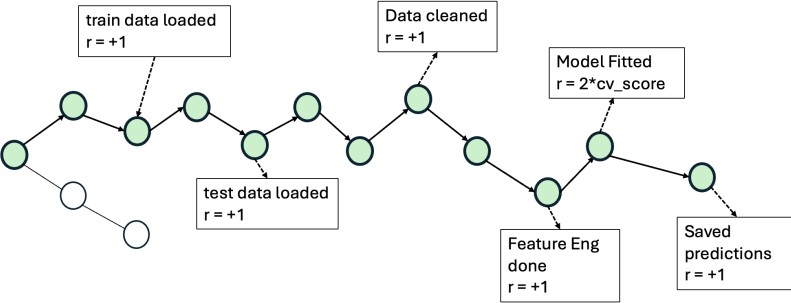

Figure 2: An example illustration of how rewards are provided in MCTS-Shaped. If a particular stage is judged to be successfully completed at a node, a reward is given, which is used to update the value of all the nodes in this trajectory. It needs to be noted that these stage-wise rewards are only provided once per trajectory and only if the earlier stages were successfully completed.

It needs to be noted that all of the stage rewards are provided to the agent only once per trajectory, and only if the earlier stages were successfully completed. The provided stage rewards are used to update the value of all the nodes in the trajectory. We verify stage completion using a reward function that inspects the node scratchpad and tool messages, confirming (i) that artifacts satisfy required properties (e.g., no NaNs for data cleaning; all columns encoded for feature engineering) and (ii) that the correct tools were invoked as evidenced by the tool logs.

## 4.5 HIERARCHICAL MCTS

We propose Hierarchical MCTS to improve over ReAct (Yao et al., 2023b), LATS (Zhou et al., 2024a), and classical MCTS (Kocsis & Szepesvari, 2006) in generating performant tool-use trajectories for solving Kaggle challenges within ML-Tool-Bench. Hierarchical MCTS decomposes a complex task into an ordered sequence of subtasks. We partition the available tools and assign them to relevant sub-tasks manually. For each subtask, MCTS searches its local state–action space to identify solution nodes. The solution nodes from one subtask are appended to the root of the next subtask, and the search continues. To avoid being trapped in locally optimal (but globally suboptimal) choices, we enumerate all solution nodes within each subtask up to a prescribed maximum subtask search depth. If there are no solution nodes identified after a subtask, the search terminates and we return 'No Solution Found'. The solution node with the highest value, at the final subtask, is returned as the solution of the Hierarchical MCTS search. Note that, when solving for each subtask in Hierarchical MCTS, we do not use any reward shaping and only check for if the subtask was solved successfully or not. Importantly, the agent is given only the tools relevant to the current sub-task (tool masking), which reduces the branching factor and focuses the search. Figure 3 illustrates the overall procedure. Hierarchical MCTS is similar to the options framework (Sutton et al., 1999), that break down a complex problem into a hierarchy of sub-tasks, making the learning process more efficient and manageable for an agent.

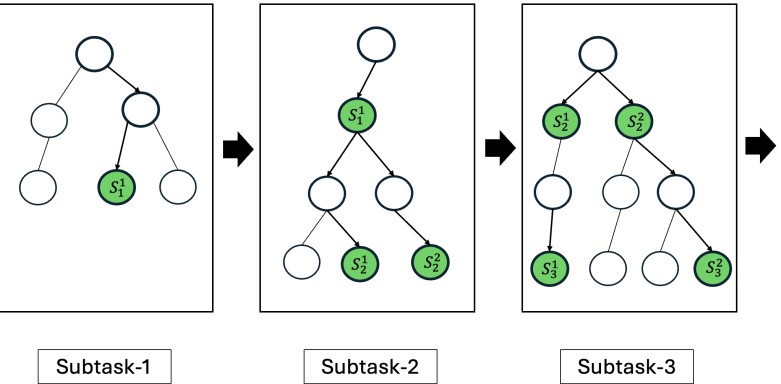

Figure 3: A schematic of Hierarchical MCTS. The task is decomposed into an ordered sequence of subtasks. For each subtask, MCTS searches for all solution nodes up to a prescribed maximum subtask depth to avoid locally optimal but globally suboptimal choices. The solution nodes from subtask $t$ are appended to the root of subtask $t+1$, and the search resumes. In the example, the solution node from subtask 1, $S_1^1$, initializes subtask 2; its solution nodes $S_2^1$ and $S_2^2$ initialize subtask 3, and so on. The highest-value solution at the final subtask is returned as the overall outcome of Hierarchical MCTS.

## 5 EXPERIMENTS

We evaluate the tool-planning performance of two language models—GPT-4o and GPT-4.1-mini, on ML-Tool-Bench. For each model, we compare five planning algorithms: (i) *ReAct* (Yao et al., 2023b); (ii) *LATS* (Zhou et al., 2024a); (iii) Monte Carlo Tree Search (MCTS) with outcome-based rewards, where the agent is rewarded upon successfully training a model or producing a valid submission file (denoted *MCTS-Outcome*); (iv) MCTS with shaped rewards, where the agent receives intermediate credit for completing stages of the Kaggle ML workflow (denoted *MCTS-Shaped*); and (v) *Hierarchical MCTS*: the Kaggle challenge is decomposed into subtasks. We use the reward stages defined for *MCTS-Shaped* as subtasks. A node is a solution node for a subtask, if it satisfies the reward condition for the stage corresponding to that subtask.

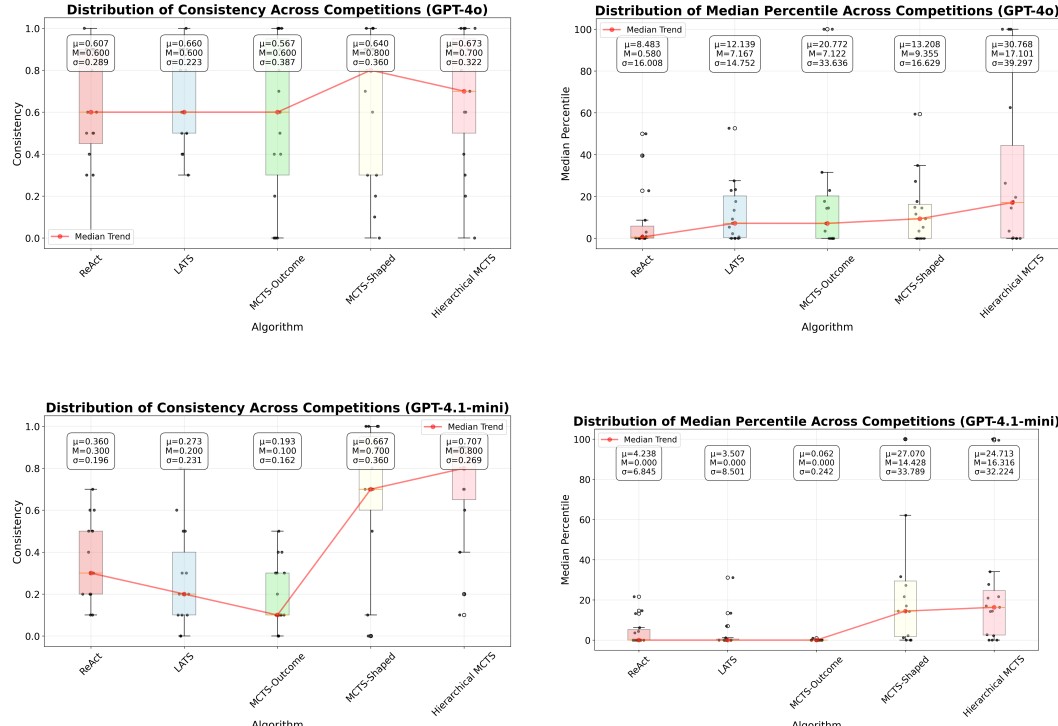

Figure 4: Plots of consistency and median leaderboard percentile across all competitions in ML-Tool-Bench, for different planning algorithms. The top row shows results for GPT-4o, with the left plot showing consistency and the right plot showing the median leaderboard percentile. The bottom row shows results for GPT-4.1-mini. Hierarchical MCTS outperforms LATS and ReAct, followed by MCTS-Shaped, in terms of leaderboard performance, for both LLMs. Also, both Hierarchical MCTS and MCTS-Shaped improve consistency over the other baselines. In the box plots, $\mu$ denotes the mean, $\sigma$ denotes the standard deviation, and $M$ denotes the Median

## 5.1 IMPLEMENTATION DETAILS

When using tree-search methods with our in-memory, named-object scheme, we adopt a *path-local scratchpad*, where each node $v$ contains a scratchpad $\mathcal{S}(v)$, that stores only the objects produced by the tool call at that node. During expansion, the LLM proposes candidate actions. For a candidate that is a tool call, the accessible memory is the path union: $\mathcal{S}^*(v) = \bigcup_{u \in \text{path}(\text{root} \to v)} \mathcal{S}(u)$, and the LLM may reference any named object in $\mathcal{S}^*(v)$ as tool arguments. The tool's outputs are written to the child's scratchpad $\mathcal{S}(\text{child})$, preserving isolation per node while enabling reuse of intermediate artifacts along the trajectory.

**LATS**: To estimate the value of a state, we provide an evaluator LLM with all `AIMessage` and `ToolMessage` entries along the path from the root to the current node; it scores the trajectory by the progress made toward solving the Kaggle challenge. To propose candidate actions, we similarly pass the full trajectory history to the LLM, which returns new reasoning steps or tool calls. Unlike the original LATS formulation, we omit a self-consistency score from the value estimate, as at each expansion the agent typically proposes a small number of distinct candidates.

**MCTS**: We propose new candidate nodes during the expansion phase using the same approach listed in LATS. To evaluate the value of a node, we check if it produces a model or a valid submission file in the outcome rewards case. In the shaped rewards case, a node is provided a reward if it successfully completes a stage, as detailed earlier. In the case of Hierarchical MCTS, we designate a node as a solution node of the subtask, if it successfully completes the stage corresponding to that subtask. Additionally, across all MCTS variants, we apply a per-level depth penalty of 0.1 to discourage unnecessarily long trajectories that fail to make progress toward the goal.

In addition to rewards, we provide targeted textual feedback to help the agent refine its plan. When a stage fails, the agent receives an explanation of the failure. For example, in *feature engineering* we flag remaining categorical columns or an excessive increase in feature dimensionality; in *data cleaning* we report the presence of missing values. If a tool invocation fails, we return an explicit message along with the tool's docstring to guide correct usage on the next attempt. We find that such feedback is crucial for consistently producing valid trajectories. This textual feedback is provided for all the MCTS variants (*MCTS-Outcome*, *MCTS-Shaped*, and *Hierarchical-MCTS*).

Ideally, we would run Monte Carlo rollouts to a fixed depth or until episode termination and use the return to update the value of all the nodes in the trajectory. Running to termination is impractical due to cost and compute constraints. Shallow rollouts (depth 3–5) are viable but GPT usage across many Kaggle challenges, planning algorithms, and trials, and roll outs at each state, resulted in extremely high costs and was infeasible. Learning value functions to approximate the value of states (Silver et al., 2016) is also not straightforward, on account of complex artifacts that are a part of the state. Consequently, we use the immediate reward at the current state (a depth-0 rollout), yielding a best-first search with a UCT-style exploration bonus. When budget permits, using small depth rollouts is preferred.

**Hierarchical MCTS:** In Hierarchical MCTS, we begin by decomposing the Kaggle challenge into a sequence of subtasks. This decomposition leverages the domain knowledge that solving a machine learning challenge typically involves data loading, data cleaning, feature engineering, modeling, evaluation, and prediction. We use a similar subtask decomposition to the reward-shaping structure used for MCTS-Shaped, described in Section 4.4.

Once we obtained textual descriptions for each subtask, a state-of-the-art coding agent was prompted with the subtask descriptions and the docstrings of the tools in our toolset, and asked to assign the relevant tools required to solve each subtask. One of the authors then manually reviewed the assignments to verify that the tool selections were sufficient and corrected minor errors made by the agent. This approach provides a general recipe for assigning tools to subtasks and can be applied in other domains, not only in the machine learning challenge–solving setting considered in this paper.

## 5.2 RESULTS

We evaluate GPT-4o and GPT-4.1-mini on our benchmark. For each algorithm–Kaggle challenge combination, we run 10 trials. We define *consistency* as the proportion of valid trajectories (e.g., 4 valid trajectories out of 10 trials yields a consistency of 0.4). For each trial, we evaluate predictions against the provided test labels using the competition's official metric and compare against the leaderboard to obtain a leaderboard percentile. For each algorithm and competition, we report the median percentile across the 10 trials. Figure 4 presents boxplots for all algorithms, summarizing the distribution of leaderboard percentiles across all competitions in our benchmark. For further details on consistency and leaderboard percentiles for both models, refer to Appendix $D$. Additionally, for details on the prompts used, refer to Appendix $H$. For additional details regarding costs, tool-masking ablations, and example trajectories refer to Appendix $G$, $F$, and $I$ respectively.

As shown in Figure 4, Hierarchical MCTS improves leaderboard performance compared to ReAct, LATS, and MCTS-Outcome, followed by MCTS-Shaped, for both GPT-4o and GPT-4.1-mini. Moreover, both Hierarchical MCTS and MCTS-Shaped achieve higher consistency than the other baselines. For GPT-4o, Hierarchical MCTS shows improvement over LATS by 9.93 percentile positions on the leaderboard and over ReAct by 16.52 percentile positions, taking the median across all competitions. For GPT-4.1-mini, Hierarchical MCTS improved over MCTS-Shaped by 1.89 percentile positions on the leaderboard, while both ReAct and LATS had a median leaderboard percentile position of 0 across all competitions. These results highlight that as toolsets become more complex and larger, it is important either to introduce hierarchy—decomposing the original task into subtasks with corresponding reward functions, or to employ shaped rewards that guide the search toward solutions. In contrast, unidirectional planning strategies like ReAct do not perform well. Similarly, tree-search methods such as LATS, that rely solely on LLM evaluation also fail, as LLMs provide inconsistent scores to nodes when trajectory lengths increase, due to the accumulation of messages and artifacts that must be considered during evaluation.

We also report the Consistency and Leaderboard percentiles (with respect to the Kaggle public leaderboard) for the five planning approaches evaluated in this paper. For this analysis, we used

the original Kaggle train and test splits rather than the smaller benchmark subsets. Since test labels were not available, we submitted our predictions to Kaggle to obtain the public leaderboard scores, which were then converted into percentile ranks. Due to cost constraints, we evaluated only a subset of six challenges from our benchmark and used GPT-4.1-mini as the underlying LLM. The results are presented in the Tables 1 and 2. We observe that these results exhibit the same trends as those seen in our benchmark evaluation: Hierarchical MCTS and MCTS-Shaped consistently outperform the other methods, while ReAct and LATS struggle, achieving a median leaderboard percentile of 0.0 across most of the six challenges evaluated.

| Competition | ReAct | LATS | MCTS-Outcome | MCTS-Shaped | Hierarchical MCTS |
|---|---|---|---|---|---|
| Spaceship Titanic | 0.3 | 0.0 | **0.6** | **0.6** | 0.4 |
| BPM Prediction | 0.2 | 0.3 | 0 | **0.8** | 0.7 |
| Calorie Expenditure Prediction | 0.4 | 0.3 | 0.2 | 0.3 | **0.6** |
| california Housing Regression | 0.6 | 0.4 | 0.1 | 0.5 | **0.8** |
| Bank Deposit Classification | 0.4 | 0.5 | 0.0 | **1.0** | 0.6 |
| Bank Churn Classification | 0.2 | 0.5 | 0.1 | 1.0 | **0.8** |
| **Overall (Median)** | 0.35 | 0.35 | 0.1 | **0.7** | 0.65 |

Table 1: Consistency scores for the five planning approaches across six Kaggle challenges, evaluated using the original train/test splits provided by Kaggle. Table added to address reviewer's questions.

| Competition | ReAct | LATS | MCTS-Outcome | MCTS-Shaped | Hierarchical MCTS |
|---|---|---|---|---|---|
| Spaceship Titanic | 0.0 | 0.0 | 39.77 | **41.69** | 0.0 |
| BPM Prediction | 0 | 0 | 0 | **5.03** | 0.19 |
| Calorie Expenditure Prediction | 0.0 | 0.0 | 0.0 | 0.0 | **16.81** |
| california Housing Regression | 5.51 | 0.0 | 0.0 | 8.99 | **24.78** |
| Bank Deposit Classification | 0.0 | 13.47 | 0.0 | **27.65** | 27.29 |
| Bank Churn Classification | 0.0 | 14.91 | 0.0 | 29.87 | **32.47** |
| **Overall (Median)** | 0.0 | 0.0 | 0.0 | 18.32 | **20.80** |

Table 2: Median leaderboard percentiles for the five planning approaches across six Kaggle challenges. Percentiles are computed from Kaggle public leaderboard scores obtained via official submissions using the original train/test splits. Table added to address reviewer's questions.

## 6 CONCLUSION

We introduced ML-Tool-Bench, a benchmark for evaluating the planning capabilities of tool-augmented LLMs on tabular Kaggle challenges. Existing tool-use benchmarks (Xu et al., 2023; Patil et al., 2025; Yao et al., 2024) primarily assess tool selection and argument grounding, rather than long-horizon planning. By contrast, many ML agents generate code directly; while flexible, this approach sacrifices modularity, reliability, and safety compared to operating within a curated toolset. Empirically, we found that ReAct and LATS struggle to consistently produce valid and performant trajectories. We proposed two improved approaches: (i) MCTS with shaped, deterministic rewards, and (ii) Hierarchical MCTS, which decomposes problems into sequenced subtasks. Across two models, Hierarchical MCTS achieved the best leaderboard performance compared to other baselines, while both Hierarchical MCTS and MCTS-Shaped improved consistency, measured as the fraction of valid trajectories. These results suggest that incorporating subtask decomposition with deterministic rewards, rather than relying on subjective LLM evaluation, yields performance gains as the set of available tools grows in size and complexity.

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

## A LARGE LANGUAGE MODEL USAGE

Large Language Models (LLMs) were used for grammatical editing and improving writing flow. Additionally, LLMs assisted the authors in conducting literature surveys and identifying related work. LLMs were also used to aid in developing the ML-Tool-Bench toolset. LLMs were used to assist with code generation, debugging, and documentation for components of the ML-Tool-Bench toolset, based on tool descriptions provided by the authors. LLMs were also used to assign relevant tools to each subtask in the proposed Hierarchical MCTS approach. All implementations were reviewed and validated by the authors. All research methodology, experimental design, data analysis, and scientific conclusions are entirely the work of the human authors.

## B APPROACHES

### B.1 MONTE CARLO TREE SEARCH

Figure 5 provides a pictorial illustration of the MCTS algorithm.

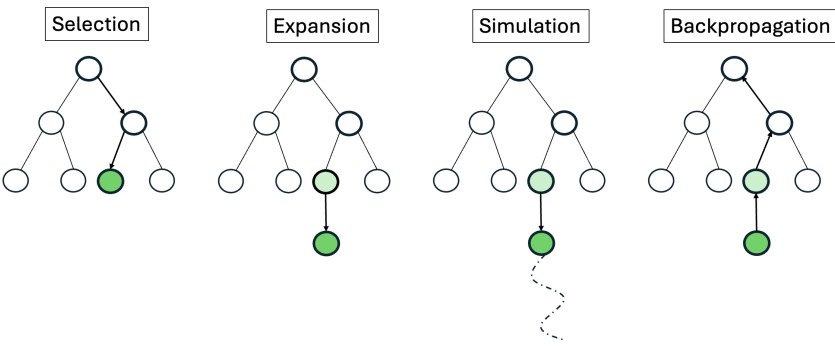

Figure 5: A pictorial illustration of Monte Carlo Tree Search

## C KAGGLE CHALLENGES

The list of Kaggle challenges present in ML-Tool-Bench, and the corresponding ML problem types of each challenge are presented in Table 3

| Challenge | Type |
|---|---|
| Santander Value Prediction Challenge | Regression |
| New York City Taxi Fare Prediction | Regression |
| New York City Taxi Trip Duration | Regression |
| Predicting the Beats-per-Minute of Songs | Regression |
| Predict Calorie Expenditure | Regression |
| Regression with a Tabular California Housing Dataset | Regression |
| Regression of Used Car Prices | Regression |
| Porto Seguro Safe Driver Prediction | Binary Classification |
| Costa Rican Household Poverty Prediction | Multi-Class Classification |
| Forest Cover Type (Kernels Only) | Multi-Class Classification |
| Santander Customer Transaction Prediction | Binary Classification |
| Binary Prediction of Poisonous Mushrooms | Binary Classification |
| Spaceship Titanic | Binary Classification |
| Binary Classification with a Bank Dataset | Binary Classification |
| Binary Classification with a Bank Churn Dataset | Binary Classification |

Table 3: Kaggle challenges used in ML-Tool-Bench with problem type.

## D RESULTS

In this section, we provide the exact consistency and performance values for each of the 15 challenges and the two models (GPT-4o and GPT-4.1-mini). Tables 4 and 5 show the consistency and leaderboard percentiles for all algorithms across all competitions in ML-Tool-Bench, for GPT-4o. Similarly, Tables 6 and 7 show the consistency and leaderboard percentiles for all algorithms across all competitions in ML-Tool-Bench, for GPT-4.1-mini.

| Competition | ReAct | LATS | MCTS-Outcome | MCTS-Shaped | Hierarchical MCTS |
|---|---|---|---|---|---|
| Spaceship Titanic | 0.6 | 0.4 | **0.9** | 0.8 | 0.6 |
| Santander Value Prediction Challenge | 0.9 | 0.6 | 0.5 | **1** | **1** |
| NYC Taxi Fare Prediction | 0 | 0.5 | 0.2 | 0.3 | **0.9** |
| NYC Taxi Trip Duration | 0.3 | 0.3 | **1** | 0.2 | 0.3 |
| BPM Prediction | **1** | 0.6 | 0 | 0.9 | 0.7 |
| Calorie Expenditure Prediction | 0.8 | 0.8 | **1** | 0.9 | **1** |
| california Housing Regression | 0.9 | 0.9 | **1** | 0.9 | 0.9 |
| Used Car Prices Regression | 0.9 | 0.4 | 0.4 | 0.9 | **1** |
| Porto Seguro Safe Driver Prediction | 0.3 | **0.5** | 0 | 0.1 | 0.2 |
| Costa Rican Household Poverty Level Prediction | 0.5 | 0.5 | 0.7 | 0.3 | **1** |
| Forest Cover Type Prediction | 0.6 | **0.9** | 0 | 0 | 0 |
| Santander Customer Transaction Prediction | 0.5 | **0.8** | **0.8** | 0.7 | 0.4 |
| Poisonous Mushroom Prediction | 0.9 | **1** | **1** | **1** | 0.8 |
| Bank Deposit Classification | 0.5 | **0.8** | 0.4 | 0.6 | 0.7 |
| Bank Churn Classification | 0.4 | 0.9 | 0.6 | **1** | 0.6 |
| **Overall (Median)** | 0.6 | 0.6 | 0.6 | **0.8** | 0.7 |

Table 4: Consistency across 15 competitions for five planning algorithms for GPT-4o.

| Competition | ReAct | LATS | MCTS-Outcome | MCTS-Shaped | Hierarchical MCTS |
|---|---|---|---|---|---|
| Spaceship Titanic | 39.54 | 0 | 22.88 | 59.44 | **62.55** |
| Santander Value Prediction Challenge | 0.09 | 7.17 | 7.12 | **14.87** | 0.27 |
| NYC Taxi Fare Prediction | 0 | 9.23 | 0 | 0 | **19.66** |
| NYC Taxi Trip Duration | 0 | 0 | **100.0** | 0 | 0 |
| BPM Prediction | 0.51 | 52.63 | 0 | 5.26 | **100** |
| Calorie Expenditure Prediction | 0.16 | 13.43 | **14.47** | **14.47** | **14.47** |
| california Housing Regression | 0.58 | 0.65 | 14.35 | 11.59 | **17.10** |
| Used Car Prices Regression | 3.0 | 0 | 0 | 9.35 | **100** |
| Porto Seguro Safe Driver Prediction | 0 | **5.34** | 0 | 0 | 0 |
| Costa Rican Household Poverty Level Prediction | 50 | 0 | **100** | 0 | **100** |
| Forest Cover Type Prediction | 0.84 | **23.32** | 0 | 0 | 0 |
| Santander Customer Transaction Prediction | 1.14 | 2.27 | **3.49** | **3.49** | 0 |
| Poisonous Mushroom Prediction | **22.75** | 17.61 | 17.69 | 17.62 | 17.62 |
| Bank Deposit Classification | 8.64 | **27.50** | 0 | 27.24 | 26.37 |
| Bank Churn Classification | 0 | 22.93 | 31.57 | **34.79** | 3.47 |
| **Overall (Median)** | 0.58 | 7.17 | 7.12 | 9.36 | **17.10** |

Table 5: Median Leaderboard percentile across 15 competitions for five planning algorithms for GPT-4o.

| Competition | ReAct | LATS | MCTS-Outcome | MCTS-Shaped | Hierarchical MCTS |
|---|---|---|---|---|---|
| Spaceship Titanic | 0.1 | 0 | 0.4 | **0.7** | 0.2 |
| Santander Value Prediction Challenge | 0.6 | 0.2 | 0 | **0.9** | **0.9** |
| NYC Taxi Fare Prediction | 0.2 | 0 | 0.1 | 0.1 | **0.9** |
| NYC Taxi Trip Duration | 0.1 | 0.1 | 0.3 | 0.5 | **0.9** |
| BPM Prediction | 0.4 | 0.1 | 0 | **1** | 0.8 |
| Calorie Expenditure Prediction | 0.6 | 0.6 | 0 | **0.9** | **0.9** |
| california Housing Regression | 0.7 | 0.3 | 0.3 | **0.8** | **0.8** |
| Used Car Prices Regression | 0.3 | 0.2 | 0.4 | **0.7** | 0.4 |
| Porto Seguro Safe Driver Prediction | 0.2 | 0.1 | 0.1 | 0.7 | **0.9** |
| Costa Rican Household Poverty Level Prediction | **0.2** | **0.2** | 0.1 | 0 | 0.1 |
| Forest Cover Type Prediction | 0.2 | 0.2 | 0.3 | 0 | **0.7** |
| Santander Customer Transaction Prediction | 0.3 | 0.5 | 0.1 | **0.7** | **0.7** |
| Poisonous Mushroom Prediction | 0.5 | 0.5 | 0.2 | **1** | 0.6 |
| Bank Deposit Classification | 0.5 | 0.3 | 0.1 | **1** | 0.9 |
| Bank Churn Classification | 0.5 | 0.8 | 0.5 | **1** | 0.9 |
| **Overall (Median)** | 0.3 | 0.2 | 0.1 | 0.7 | **0.8** |

Table 6: Consistency across 15 competitions for five planning algorithms for GPT-4.1-mini.

| Competition | ReAct | LATS | MCTS-Outcome | MCTS-Shaped | Hierarchical MCTS |
|---|---|---|---|---|---|
| Spaceship Titanic | 0 | 0 | 0 | **62.11** | 0 |
| Santander Value Prediction Challenge | 3.60 | 0 | 0 | **14.24** | **14.24** |
| NYC Taxi Fare Prediction | 0 | 0 | 0 | 0 | **20.94** |
| NYC Taxi Trip Duration | 0 | 0 | 0 | 1.24 | **2.70** |
| BPM Prediction | 0 | 0 | 0 | **100** | **100** |
| Calorie Expenditure Prediction | 4.31 | 13.41 | 0 | **14.43** | **14.43** |
| california Housing Regression | **21.59** | 0 | 0 | **21.59** | **21.59** |
| Used Car Prices Regression | 0 | 0 | 0 | **100** | 0 |
| Porto Seguro Safe Driver Prediction | 0 | 0 | 0 | **17.01** | **17.01** |
| Costa Rican Household Poverty Level Prediction | 0 | 0 | 0 | 0 | 0 |
| Forest Cover Type Prediction | 0 | 0 | 0 | 0 | **99.44** |
| Santander Customer Transaction Prediction | 0 | 1.17 | 0 | **2.27** | **2.27** |
| Poisonous Mushroom Prediction | 6.23 | 6.97 | 0 | 14.36 | **16.32** |
| Bank Deposit Classification | 13.19 | 0 | 0 | 27.24 | **27.73** |
| Bank Churn Classification | 14.66 | 31.09 | 0.94 | 31.57 | **34.02** |
| **Overall (Median)** | 0 | 0 | 0 | 14.43 | **16.32** |

Table 7: Median Leaderboard percentile across 15 competitions for five planning algorithms for GPT-4.1-mini.

| Stage | Number of Tools |
|---|---|
| Data Loading | 6 |
| Data Cleaning | 9 |
| Feature Engineering | 30 |
| Modeling | 10 |
| Evaluation/Prediction | 10 |

Table 8: Number of tools available at each stage of a Kaggle-style workflow. In total, 61 tools are provided spanning data loading, cleaning, feature engineering, and modeling. Some tools can appear in more than one stage

# E  TOOLS

In this section, we describe the various tools that are part of ML-Tool-Bench. Table 8 shows the number of tools in our toolset that are part of each stage in solving an ML challenge on Kaggle. Table 9 provides info about all the tools in the curated toolset provided by ML-Tool-Bench

**Decorators for named references**  To enable tools to operate on named references rather than raw objects, we design four decorators that adapt arbitrary user-provided functions to our scratchpad interface according to their read–write behavior. We categorize tools into four types:

1. **Set tool**: saves an object to memory. Example: `read_csv` loads a dataframe and stores it under a provided name.

2. **Get tool**: reads an object from memory. Example: `get_dataframe_summary` loads a dataframe and returns a brief textual summary to guide subsequent planning.

3. **Get–Set tool**: reads an object from memory and writes a new object to memory. Example: `fit_randomforest_model` takes as input, a dataframe, and returns a fitted model.

4. **Override tool**: reads an object, returns an updated object, and overwrites the input variable binding with the returned value. Example: `cast_column` loads a dataframe and returns a modified dataframe that replaces the original.

Accordingly, we provide four decorators: `make_get_tool`, `make_set_tool`, `make_get_and_set_tool`, and `make_override_tool`, that automatically wrap user-provided tools to operate on named references and integrate with the scratchpad.

| Function Signature | Description |
|---|---|
| **Modeling Functions** | |

| `fit_logistic_regressor(X_train, y_train, cv=5)` | Fit Logistic Regression model |
|---|---|
| `fit_linear_regressor(X_train, y_train, cv=5)` | Fit Linear Regression model |
| `fit_random_forest_regressor(X_train, y_train, cv=5)` | Fit Random Forest Regressor |
| `fit_random_forest_classifier(X_train, y_train, cv=5)` | Fit Random Forest Classifier |
| `fit_xgboost_regressor(X_train, y_train, cv=5)` | Fit XGBoost Regressor |
| `fit_xgboost_classifier(X_train, y_train, cv=5)` | Fit XGBoost Classifier |
| `fit_lightgbm_regressor(X_train, y_train, cv=5)` | Fit LightGBM Regressor |
| `fit_lightgbm_classifier(X_train, y_train, cv=5)` | Fit LightGBM Classifier |
| `fit_catboost_regressor(X_train, y_train, cv=5)` | Fit CatBoost Regressor |
| `fit_catboost_classifier(X_train, y_train, cv=5)` | Fit CatBoost Classifier |
| **Data Loading Functions** | |
| `read_data(filepath)` | Read CSV data into a pandas DataFrame |
| **Feature Engineering and Functions to get Dataframe information** | |
| `create_numeric_feature(df, name, expression)` | Create a numeric feature using a pandas expression |
| `create_categorical_feature(df, name, source_column, mapping)` | Create a categorical feature by mapping values from a source column |
| `create_conditional_feature(df, name, condition, true_value, false_value)` | Create a feature based on a condition |
| `extract_string_pattern(df, name, source_column, pattern, group=0)` | Extract pattern from string column using regex |
| `split_string_column(df, name_prefix, source_column, delimiter, max_splits=-1, indices=None)` | Split string column and create separate features |
| `create_group_aggregation(df, name, group_column, agg_column, agg_func)` | Create feature by aggregating within groups |
| `get_group_aggregation(df, group_column, agg_column, agg_func)` | Get aggregation result without adding it to the DataFrame |
| `create_rolling_feature(df, name, source_column, window, agg_func='mean')` | Create rolling window feature |
| `create_lag_feature(df, name, source_column, lag=1)` | Create lagged feature |
| `create_lead_feature(df, name, source_column, lead=1)` | Create leading feature |
| `extract_datetime_features(df, datetime_column, features=None)` | Extract datetime features from datetime column |
| `create_time_delta(df, name, start_column, end_column, unit='D')` | Create time delta feature between two datetime columns |
| `apply_custom_function(df, name, source_columns, func)` | Apply custom function to create feature |
| `fillna_with_value(df, columns, value)` | Fill missing values with a specific value |
| `fillna_with_median(df, columns=None)` | Fill missing values with median of the column |
| `fillna_with_mean(df, columns=None)` | Fill missing values with mean of the column |

| `fillna_with_mode(df, columns=None)` | Fill missing values with mode of the column |
|---|---|
| `fillna_with_condition(df, target_column, condition, fill_value)` | Fill missing values in a column based on a condition |
| `fillna_with_multiple_conditions(df, target_column, conditions_and_values)` | Fill missing values in a column based on multiple conditions |
| `fillna_with_conditional_aggregation(df, target_column, condition_column, condition_values, agg_func='mean')` | Fill missing values using conditional aggregation based on another column's values |
| `fillna_with_custom_function(df, target_column, condition, custom_func)` | Fill missing values using a custom function based on a condition |
| `drop_rows_with_missing(df, columns=None, threshold=None)` | Drop rows with missing values |
| `get_missing_summary(df)` | Get a summary of missing values in the DataFrame |
| `cast_columns(df, column_type_mapping)` | Cast columns to specified data types |
| `cast_numeric_columns(df, columns=None, target_type='float')` | Cast numeric columns to specified type |
| `cast_integer_columns_to_float(df, columns=None)` | Cast integer columns to float type |
| `cast_categorical_columns(df, columns=None)` | Cast categorical columns to category type |
| `one_hot_encode(df, columns=None, drop_first=True, prefix=None)` | One-hot encode categorical columns |
| `label_encode(df, columns=None)` | Label encode categorical columns |
| `normalize_features(df, columns=None, method='standard')` | Normalize numeric features |
| `encode_all_categorical_columns(df, method='one_hot', drop_first=True)` | Encode all categorical/object columns using specified method |
| `normalize_all_numerical_columns(df, method='standard')` | Normalize all numerical columns using specified method |
| `concatenate_train_test(train_df, test_df)` | Concatenate train and test data with tracking columns for proper splitting |
| `split_combined_into_train_test(combined)` | Split combined data back into train and test using tracking columns |
| `convert_dataframe_to_features_target(df, target_column, is_train=True)` | Convert DataFrame to features and target format |
| `convert_to_dataframe(data, **kwargs)` | Convert various data types to pandas DataFrame |
| `drop_feature(df, column)` | Drop feature(s) from the DataFrame |
| `get_features(df, columns)` | Extract specific features (columns) from the DataFrame |
| `concatenate_dataframes(df1, df2, axis=0)` | Concatenate two DataFrames |
| `join_dataframes(left_df, right_df, left_on, right_on=None, how='inner', suffixes=('_x', '_y'))` | Join two DataFrames using pandas merge functionality |
| `rename_feature(df, old_name, new_name)` | Rename feature(s) |
| `get_unique_values(df, column, sort=True, include_counts=True)` | Get unique values from a column as a DataFrame |
| `get_dataframe_dtypes_summary(df)` | Get comprehensive summary of the dtypes in the entire DataFrame |
| `filter_dataframe(df, condition)` | Filter DataFrame using a boolean condition |
| **Model Utilities** | |
| `save_model(model, filepath='model.pkl')` | Save the trained model to disk using pickle |

| `load_model(filepath)` | Load a trained model from disk using pickle |
|---|---|
| `save_dataframe_to_csv(df, filepath)` | Save a DataFrame to CSV file |
| **Model Evaluation Functions** | |
| `evaluate_regression_model(model, X_test, y_test, model_name="model", eval_data_label='test')` | Evaluate a trained regression model on data |
| `evaluate_classification_model(model, X_test, y_test, model_name="model", eval_data_label='test')` | Evaluate a trained classification model on data |
| `predict_target(model, X_data, model_name="model", return_probabilities=False)` | Make predictions using a trained model |

Table 9: All tools in the curated toolset provided by ML-Tool-Bench.

## F   TOOL MASKING ABLATIONS

In this section, we perform an ablation study to investigate if tool masking contributes significantly to the performance of Hierarchical MCTS. We select a subset of five Kaggle challenges from our benchmark and evaluate all of the planning approaches, alongside a Hierarchical MCTS approach that does not use tool masking, i.e all tools are available to the agent during all the subtasks. We use GPT-4.1-mini for our experiments instead of GPT-4o for cost reasons. The results are presented in Tables 10 and 11. The results demonstrate that the performance of Hierarchical MCTS degrades substantially without tool masking. Hierarchical MCTS without tool masking, achieves a median consistency of 0.3 and a median leaderboard percentile position of 0 across the chosen subset of five challenges. In comparison, Hierarchical MCTS with tool masking achieves a median consistency and leaderboard percentile of 0.8 and 21.10 respectively. This highlights that both tool masking and subtask decomposition are critical for effectively solving long-horizon planning problems in high-dimensional action spaces using LLMs.

| Competition | ReAct | LATS-Reflection | MCTS-Outcome | MCTS-Shaped | Hierarchical MCTS | Hierarchical MCTS (No Tool Masking) |
|---|---|---|---|---|---|---|
| Spaceship Titanic | 0.1 | 0.1 | 0.1 | **_0.7_** | 0.5 | 0 |
| Poisonous Mushroom Prediction | 0.4 | 0.5 | 0.2 | **_0.6_** | 0.3 | 0.2 |
| Bank Churn Classification | 0.3 | 0.3 | 0.3 | **_0.9_** | **_0.9_** | 0.3 |
| Santander Customer Transaction Prediction | 0.6 | 0.2 | 0.0 | 0.7 | **_0.9_** | 0.3 |
| NYC Taxi Fare Prediction | 0.0 | 0.0 | 0.4 | 0.1 | **_0.8_** | 0.4 |
| **Overall (Median)** | 0.3 | 0.2 | 0.2 | 0.7 | **_0.8_** | 0.3 |

Table 10: Consistency across five competitions for GPT-4.1-mini with six planning algorithm variants. Maximum values per row are highlighted. The results demonstrate that the performance of Hierarchical MCTS degrades substantially without tool masking.

## G   COST COMPARISIONS

We also provide cost comparisons for all planning algorithms using GPT-4.1-mini on the same subset of five Kaggle challenges from our benchmark that was used in the Tool Ablation Study (Appendix F). The results are reported in Table 12. LATS is the most expensive planning approach, costing 3.5× more than the more successful variants (Hierarchical-MCTS and MCTS-Shaped), while achieving only a consistency of 0.2 and a median percentile position of 0 across the five Kaggle challenges. This suggests that LATS's search is unfocused and tends to wander due to inconsistent scoring

| Competition | ReAct | LATS-Reflection | MCTS-Outcome | MCTS-Shaped | Hierarchical MCTS | Hierarchical MCTS (No Tool Masking) |
|---|---|---|---|---|---|---|
| Spaceship Titanic | 0.0 | 0.0 | 0.0 | **53.59** | 28.01 | 0.0 |
| Poisonous Mushroom Prediction | 0.0 | 6.60 | 0.0 | **16.99** | 0.0 | 0.0 |
| Bank Churn Classification | 0.0 | 0.0 | 0.0 | **31.57** | **31.57** | 0.0 |
| Santander Customer Transaction Prediction | **2.27** | 0.0 | 0.0 | **2.27** | **2.27** | 0.0 |
| NYC Taxi Fare Prediction | 0.0 | 0.0 | 0.0 | 0.0 | **21.10** | 0.0 |
| **Overall (Median)** | 0.0 | 0.0 | 0.0 | 16.99 | **21.10** | 0.0 |

Table 11: Median leaderboard percentile across five competitions for GPT-4.1-mini with six planning algorithm variants. Maximum values per row are highlighted. The results demonstrate that the performance of Hierarchical MCTS degrades substantially without tool masking.

by the LLM evaluator. ReAct is the cheapest method but also performs poorly, only marginally outperforming LATS despite the latter using 10.5× more budget.

| Competition | ReAct | LATS-Reflection | MCTS-Outcome | MCTS-Shaped | Hierarchical MCTS |
|---|---|---|---|---|---|
| Spaceship Titanic | 1.66 | **22.29** | 3.22 | 8.44 | 3.93 |
| Poisonous Mushroom Prediction | 0.86 | **13.64** | 3.1.3 | 2.86 | 1.89 |
| Bank Churn Classification | 1.21 | **9.98** | 1.44 | 1.79 | 5.65 |
| Santander Customer Transaction Prediction | 1.34 | **11.28** | 2.36 | 1.43 | 5.38 |
| NYC Taxi Fare Prediction | 2.02 | **17.83** | 2.72 | 4.07 | 5.56 |
| **Overall (Sum)** | 7.08 | **75.02** | 12.86 | 18.59 | 22.42 |

Table 12: Total costs ($) aggregated over 10 trajectories for each of the five competitions using GPT-4.1-mini under all planning algorithm variants examined in this study. For each row, the maximum value is highlighted

# H   PROMPTS

For each competition in our benchmark, we constructed a standardized instruction template that included: (i) a brief description of the Kaggle challenge, (ii) a description of the data fields, and (iii) additional requirements specified from our end regarding model training and submission. For the challenge and data field descriptions, we used only the information provided in the corresponding sections of the original Kaggle competition; no human-authored modifications or additions were introduced.

Below, we show the exact template used for the *Spaceship Titanic* challenge (Howard et al., 2022):

> Welcome to the year 2912, where your data science skills are needed to solve a cosmic mystery. We've received a transmission from four lightyears away and things aren't looking good.
>
> The Spaceship Titanic was an interstellar passenger liner launched a month ago. With almost 13,000 passengers on board, the vessel set out on its maiden voyage transporting emigrants from our solar system to three newly habitable exoplanets orbiting nearby stars.
>
> While rounding Alpha Centauri en route to its first destination—the torrid 55 Cancri E—the unwary Spaceship Titanic collided with a spacetime anomaly hidden within a dust cloud. Sadly, it met a similar fate as its namesake from 1000 years before. Though the ship stayed intact, almost half of the passengers were transported to an alternate dimension! Read the data from the train.csv file in the data folder.

To help rescue crews and retrieve the lost passengers, you are challenged to predict which passengers were transported by the anomaly using records recovered from the spaceship's damaged computer system.

In this competition your task is to predict whether a passenger was transported to an alternate dimension during the Spaceship Titanic's collision with the space-time anomaly. To help you make these predictions, you're given a set of personal records recovered from the ship's damaged computer system.

**File and Data Field Descriptions**

- `PassengerId` – Unique identifier in the form gggg_pp, where gggg indicates a group and pp a member index.
- `HomePlanet` – Planet of permanent residence.
- `CryoSleep` – Whether the passenger elected suspended animation.
- `Cabin` – Cabin number, formatted as deck/num/side.
- `Destination` – Planet of disembarkation.
- `Age` – Age of passenger.
- `VIP` – Whether the passenger paid for VIP services.
- `RoomService`, `FoodCourt`, `ShoppingMall`, `Spa`, `VRDeck` – Spending at onboard amenities.
- `Name` – Full passenger name.
- `Transported` – Target variable — whether the passenger was transported.

**Submission File Format**

- `PassengerId` – Identifier for each passenger in the test set.
- `Transported` – Prediction (True/False).

**Benchmark Instructions**

- The training data is located at `data/spaceship_titanic/train.csv`.
- The test data is located at `data/spaceship_titanic/test.csv`.
- Load, clean, and perform feature engineering before fitting models.
- Concatenate train and test datasets before preprocessing to ensure consistent transformations, then split back.
- Experiment with multiple models and hyperparameter tuning to find the best-performing solution.
- Report evaluation results demonstrating model fit.
- Save the best model to `model_saves/spaceship_titanic/`.
- Save predictions for the test set in CSV format to `{save_directory}/{save_file_name}.csv`.

## H.1 LATS

For LATS (Zhou et al., 2024a), we used the following system prompt for the reflection step, where an LLM was asked to evaluate the quality of a given trajectory (i.e., all message contents of the trajectory were passed as input):

> You are a Data Science judge, who evaluates the goodness of tool calling trajectories to solve Machine Learning tasks on Kaggle. Reflect and grade the agent's trajectory plan for the provided challenge. The trajectories should be aimed towards solving the challenge, i.e., generating a trained model and a valid submission file. Keep your reflections concise and to the point.

For the expansion stage, we used the following system prompt. The trajectory of messages up to the current node was passed as input, and the agent was asked to propose new expansion candidates:

> You are a Data Scientist tasked with solving the Kaggle competition provided below, with the tools available to you. Propose tool candidates that would help solve the problem at the current stage.

## H.2   MCTS

For both MCTS-Outcome and MCTS-Shaped, we used the same system prompt in the expansion stage, as was used for LATS. The trajectory of messages up to the current node was passed as input, and the agent was asked to propose new expansion candidates (same as LATS)

## H.3   Hierarchical MCTS

For Hierarchical-MCTS, to propose candidates during the expansion phase, we used subtask-specific prefixes for the system prompt. Each subtask was associated with a descriptive prefix that constrained the role of the agent and defined the completion condition for that stage. The list of prefixes is shown below:

- `train_data_loading`: "You are a Data Scientist in the Data Loading stage of solving a Kaggle challenge, using only the tools available to you. This stage ends when you have loaded the train data successfully."

- `test_data_loading`: "You are a Data Scientist in the Data Loading stage of solving a Kaggle challenge, using only the tools available to you. This stage ends when you have loaded the test data successfully."

- `combine_train_test`: "You are a Data Scientist in the Data Loading stage of solving a Kaggle challenge, using only the tools available to you. This stage ends when you have combined the train and test data into a single dataframe successfully, to be used for downstream Data Cleaning and Feature Engineering."

- `data_cleaning`: "You are a Data Scientist in the Data Cleaning stage of solving a Kaggle challenge, using only the tools available to you. This stage ends when there are no missing values present in the data. This also includes the column corresponding to the target variable, that may have NaNs in the test partition since the target variable is not present in the test partition. You are allowed to be innovative in filling the missing values based on your understanding of the data."

- `feature_engineering`: "You are a Data Scientist in the Feature Engineering stage of solving a Kaggle challenge, using only the tools available to you. Create new features, or delete unimportant features or transform existing features as needed. You are not allowed to delete or modify features that indicate if the row in the data belongs to the train or test partition. You are also not allowed to augment the feature corresponding to the target variable. Use your understanding of the data to aid your decisions. This stage ends when the models feel that the features are good enough for modeling, and categorical and numerical features have been properly encoded. After the end of this stage, all the features should be (i) either int or float or (ii) int, float, category with the number of unique values in the category columns not being exorbitantly large."

- `split_train_test`: "You are a Data Scientist in the Split Train Test stage of solving a Kaggle challenge, using only the tools available to you. Split the combined train and test data into train and test dataframes. This stage ends when the train and test dataframes are successfully split from the combined dataframe."

- `train_data_to_features_target`: "You are a Data Scientist in the Converting the Train Data to Features and Target stage of solving a Kaggle challenge, using only the tools available to you. Convert the train data into features and target. This stage ends when the train data is successfully converted into features and target, for making downstream modeling upon."

- `test_data_to_features`: "You are a Data Scientist in the Converting the Test Data to Features stage of solving a Kaggle challenge, using only the tools available to you. Convert the test data into features. This stage ends when the test data is successfully converted into features, for making downstream predictions upon."

- `modeling`: "You are a Data Scientist in the Modeling stage of solving a Kaggle challenge, using only the tools available to you. Train and tune models. You might need to experiment with different model choices and properly tune your hyperparameters to get good performance. Use the provided evaluation tools to evaluate your trained models if

needed. This stage ends when the agent has successfully created a model that it considers to be the best."

- `create_submission_dataframe`: "You are a Data Scientist in the Create Submission stage of solving a Kaggle challenge, using only the tools available to you. Make predictions on the test data, and create a submission dataframe that contains the predictions in the requested format. This stage ends when the submission dataframe in the correct format is created."

The system prompt was then constructed as:

{`subtask_description`}. Propose tool candidates that would help solve the problem at the current stage.

The trajectory of messages up to the current node was passed as input, and the agent was asked to propose new expansion candidates (same as what was used for LATS, MCTS variants)

## I  VISUALIZING TRAJECTORIES

In this section, we present example trajectories for each of the planning algorithms evaluated in this paper. For every algorithm, we provide both a successful and a failed trajectory. All trajectories are collected on the Spaceship Titanic Kaggle challenge using the GPT-4.1-mini model, ensuring a consistent evaluation setting across planning methods.

### I.1  REACT

#### I.1.1  FAILED TRAJECTORY

The following failed ReAct trajectory contains 12 tool calls and terminates during the Feature Engineering stage, never reaching the Modeling stage. This pattern is common among failed ReAct runs: a substantial fraction are unable to complete the full end-to-end machine learning pipeline required for the Kaggle task. This snippet corresponds to the final portion of the failed ReAct trajectory.

```
{
  "step_number": 20,
  "timestamp": "2025-11-20T18:52:06.565366",
  "step_type": "tool_execution_initiation",
  "action": "tool_execution_started",
  "tools_to_execute": [
    {
      "tool_name": "apply_custom_function",
      "tool_args": {
        "bindings": { "df": "combined_data" },
        "output": "combined_data",
        "func_kwargs": {
          "name": "Cabin_deck",
          "source_columns": "Cabin",
          "func": "lambda x: x.split('/')[0] if isinstance(x, str) else
  None"
        }
      },
      "tool_id": "call_s5ZjVgGT9j6llr3Vg7jUQavA"
    }
  ]
},
{
  "step_number": 21,
  "timestamp": "2025-11-20T18:52:06.599497",
  "step_type": "tool_execution_completion",
  "action": "tool_execution_completed",
  "tool_results": [
    {
```

```
        "message_type": "ToolMessage",
        "content_preview": "Error: TypeError(\"'str' object is not
    callable\")\n Please fix your mistakes.",
        "content_length": 75
      }
    ]
  },
  {
    "step_number": 22,
    "timestamp": "2025-11-20T18:52:10.291750",
    "step_type": "tool_execution_initiation",
    "action": "tool_execution_started",
    "tools_to_execute": [
      {
        "tool_name": "apply_custom_function",
        "tool_args": {
          "bindings": { "df": "combined_data" },
          "output": "combined_data",
          "func_kwargs": {
            "name": "Cabin_deck",
            "source_columns": "Cabin",
            "func": null
          }
        },
        "tool_id": "call_JiDjhYHW16xgodP4y69hOAR3"
      }
    ]
  },
  {
    "step_number": 23,
    "timestamp": "2025-11-20T18:52:10.324847",
    "step_type": "tool_execution_completion",
    "action": "tool_execution_completed",
    "tool_results": [
      {
        "message_type": "ToolMessage",
        "content_preview": "Error: TypeError(\"'NoneType' object is not
    callable\")\n Please fix your mistakes.",
        "content_length": 80
      }
    ]
  }
```

### I.1.2 SUCCESSFUL TRAJECTORY

This trajectory comprises 40 tool calls and demonstrates strong performance. The feature engineering stage involves numerous successful and relevant tool invocations. In contrast, the modeling stage fits only a CatBoost classifier without exploring additional model families. When scored on our internal test set, this trajectory attains a percentile of 54.92 on the Kaggle leaderboard for this task.

```
  {
    "step_number": 70,
    "timestamp": "2025-11-20T18:53:09.461595",
    "step_type": "tool_execution_initiation",
    "action": "tool_execution_started",
    "tools_to_execute": [
      {
        "tool_name": "convert_dataframe_to_features_target",
        "tool_args": {
          "bindings": {
            "df": "test_df"
          },
          "func_kwargs": {
```

```
                  "target_column": "Transported",
                  "is_train": false
                },
                "output": "test_features_target"
              },
              "tool_id": "call_VTa2DIpTuPJRLydhruqaaiBK"
            }
          ]
        },
        {
          "step_number": 71,
          "timestamp": "2025-11-20T18:53:09.535408",
          "step_type": "tool_execution_completion",
          "action": "tool_execution_completed",
          "tool_results": [
            {
              "message_type": "ToolMessage",
              "content_preview": "Applied convert_dataframe_to_features..
              "content_length": 864
            }
          ]
        },
        {
          "step_number": 72,
          "timestamp": "2025-11-20T18:53:11.188346",
          "step_type": "tool_execution_initiation",
          "action": "tool_execution_started",
          "tools_to_execute": [
            {
              "tool_name": "tune_catboost_classifier",
              "tool_args": {
                "bindings": {
                  "X_train": "X_train",
                  "y_train": "Y_train"
                },
                "output": "catboost_tuning_result"
              },
              "tool_id": "call_DHeRtQVO4RZb8ayZyc47Ow0t"
            }
          ]
        },
        {
          "step_number": 73,
          "timestamp": "2025-11-20T18:53:12.310197",
          "step_type": "tool_execution_completion",
          "action": "tool_execution_completed",
          "tool_results": [
            {
              "message_type": "ToolMessage",
              "content_preview": "Applied tune_catboost_classifier. The Best
        params and CV score for this method are {'depth': 6, 'iterations':
        50, 'l2_leaf_reg': 1, 'learning_rate': 0.1} and 0.8011213918872091
        respectively. The mapping between the function parameters and the
        scratchpad k... [truncated]",
              "content_length": 1088
            }
          ]
        },
        {
          "step_number": 74,
          "timestamp": "2025-11-20T18:53:14.269490",
          "step_type": "tool_execution_initiation",
          "action": "tool_execution_started",
          "tools_to_execute": [
            {
```

```
              "tool_name": "evaluate_classification_model",
              "tool_args": {
                "bindings": {
                  "model": "best_estimator",
                  "X_test": "X_train",
                  "y_test": "Y_train"
                },
                "func_kwargs": {
                  "model_name": "CatBoost Classifier",
                  "eval_data_label": "train"
                }
              },
              "tool_id": "call_76EXOjjGyUp86wMwe3bVbES4"
          }
        ]
      },
      {
        "step_number": 75,
        "timestamp": "2025-11-20T18:53:14.363633",
        "step_type": "tool_execution_completion",
        "action": "tool_execution_completed",
        "tool_results": [
          {
            "message_type": "ToolMessage",
            "content_preview": "Applied evaluate_classification_model..
    [truncated]",
            "content_length": 1753
          }
        ]
      },
      {
        "step_number": 76,
        "timestamp": "2025-11-20T18:53:16.071819",
        "step_type": "tool_execution_initiation",
        "action": "tool_execution_started",
        "tools_to_execute": [
          {
            "tool_name": "predict_target",
            "tool_args": {
              "bindings": {
                "model": "best_estimator",
                "X_data": "X_test"
              },
              "func_kwargs": {
                "model_name": "CatBoost Classifier"
              },
              "output": "test_predictions"
            },
            "tool_id": "call_fM4PS3vRAthZWoDZw6ycGzUQ"
          }
        ]
      },
      {
        "step_number": 77,
        "timestamp": "2025-11-20T18:53:16.166858",
        "step_type": "tool_execution_completion",
        "action": "tool_execution_completed",
        "tool_results": [
          {
            "message_type": "ToolMessage",
            "content_preview": "Applied predict_target ..[truncated]",
            "content_length": 1019
          }
        ]
      },
```

```
     {
       "step_number": 78,
       "timestamp": "2025-11-20T18:53:26.236014",
       "step_type": "tool_execution_initiation",
       "action": "tool_execution_started",
       "tools_to_execute": [
         {
           "tool_name": "save_model",
           "tool_args": {
             "bindings": {
               "model": "best_estimator"
             },
             "func_kwargs": {
               "filepath":
     "model_saves/spaceship_titanic/catboost_best_model.pkl"
             }
           },
           "tool_id": "call_PI0hV7OnLQEBSrir8PNgWSU0"
         }
       ]
     },
     {
       "step_number": 79,
       "timestamp": "2025-11-20T18:53:26.322763",
       "step_type": "tool_execution_completion",
       "action": "tool_execution_completed",
       "tool_results": [
         {
           "message_type": "ToolMessage",
           "content_preview": "Applied save_model ..",
           "content_length": 676
         }
       ]
     },
     {
       "step_number": 80,
       "timestamp": "2025-11-20T18:53:29.230047",
       "step_type": "tool_execution_initiation",
       "action": "tool_execution_started",
       "tools_to_execute": [
         {
           "tool_name": "save_dataframe_to_csv",
           "tool_args": {
             "bindings": {
               "df": "test_predictions"
             },
             "func_kwargs": {
               "filepath":
     "/submissions/spaceship_titanic/react/react_agent_4_47a197f8-edb6-4c43-a9ce-ae5f8ef45736.c
             }
           },
           "tool_id": "call_xrfZg9Pg0FMXxgpr7a2rAGxf"
         }
       ]
     },
     {
       "step_number": 81,
       "timestamp": "2025-11-20T18:53:29.329762",
       "step_type": "tool_execution_completion",
       "action": "tool_execution_completed",
       "tool_results": [
         {
           "message_type": "ToolMessage",
           "content_preview": "Applied save_dataframe_to_csv..",
           "content_length": 777
```

```
1458                }
1459            ]
1460        },
1461        {
1462            "step_number": 82,
1463            "timestamp": "2025-11-20T18:53:34.098547",
1464            "step_type": "execution_summary",
1465            "action": "agent_execution_completed",
1466            "total_execution_time": 114.67485404014587,
1467            "total_tokens": 1022157,
1468            "total_cost": 0.156462,
1469            "final_message_count": 81,
1470            "competition_name": "spaceship_titanic"
1471        }
1472    ]
1473  }
```

## I.2 LATS

## I.3 FAILED TRAJECTORY

This trajectory contains 17 tool calls. Although this trajectory, which denotes the agent's most promising branch, successfully fit a model, LATS ultimately prioritized other failed branches, causing it to miss this viable solution path. As a result, the final output did not include any valid trajectory that solved the challenge. This failure stems from inconsistencies in the reflection-based rating process, which incorrectly ranked failed nodes above successful progress.

```
    {
        "step_number": 37,
        "message_type": "AIMessage",
        "timestamp": "2025-11-21T00:42:26.668798",
        "step_type": "tool_selection",
        "action": "selected_tools_for_execution",
        "tools_selected": [
            "tune_lightgbm_classifier"
        ],
        "tool_calls_detail": [
            {
                "name": "tune_lightgbm_classifier",
                "args": {
                    "bindings": {
                        "X_train": "X_train",
                        "y_train": "Y_train"
                    },
                    "output": "best_lgbm_model"
                },
                "id": "call_iff8Jho9Tc91a1v1bEPfxIsZ",
                "type": "tool_call"
            }
        ],
        "content": "Features and target variables are now separated. I
will proceed with hyperparameter tuning for a classification model
to find the best model for this task. I will start with a LightGBM
Classifier tuned using cross-validation."
    },
    {
        "step_number": 38,
        "message_type": "ToolMessage",
        "timestamp": "2025-11-21T00:42:26.668802",
        "step_type": "tool_result",
        "action": "received_tool_output",
        "content_preview": "Error: ValueError('Input y contains NaN.')\n
Please fix your mistakes.",
        "content_length": 69
```

```
        },
        {
          "step_number": 39,
          "message_type": "HumanMessage",
          "timestamp": "2025-11-21T00:42:26.668807",
          "step_type": "reflection",
          "action": "llm_reflection",
          "content_preview": "Reasoning: The agent failed to check for and
      handle missing target values before model tuning, resulting in a
      ValueError due to NaNs in the target variable. This indicates a gap
      in data cleaning and validation steps before model training. Proper
      imputation or removal of missing target rows is necess...",
          "content_length": 704,
          "extracted_score": 4.0,
          "full_reflection_content": "Reasoning: The agent failed to check
      for and handle missing target values before model tuning, resulting
      in a ValueError due to NaNs in the target variable. This indicates a
      gap in data cleaning and validation steps before model training.
      Proper imputation or removal of missing target rows is necessary.
      The agent should have performed or verified missing value handling
      comprehensively before tuning models. This oversight blocks progress
      in model fitting and prediction generation, which are critical to
      solving the task. The trajectory is incomplete and insufficient as
      no valid model training was achieved due to missing data issues.
      Addressing this gap is essential for a correct solution.\nScore: 4"
        },
        {
          "step_number": 40,
          "message_type": "AIMessage",
          "timestamp": "2025-11-21T00:42:26.668821",
          "step_type": "tool_selection",
          "action": "selected_tools_for_execution",
          "tools_selected": [
            "filter_dataframe"
          ],
          "tool_calls_detail": [
            {
              "name": "filter_dataframe",
              "args": {
                "bindings": {
                  "df": "combined_df"
                },
                "output": "combined_train_df",
                "func_kwargs": {
                  "condition": "Transported.notna()"
                }
              },
              "id": "call_seR8LUmqkK389bcayn0QbxWI",
              "type": "tool_call"
            }
          ],
          "content": "The model tuning failed because there are missing
      values in the target column \"Transported\". I need to clean the
      data by removing rows with missing target values before proceeding
      with model training. I will filter out such rows from the combined
      dataset and then separate features and target again."
        },
        {
          "step_number": 41,
          "message_type": "ToolMessage",
          "timestamp": "2025-11-21T00:42:26.668824",
          "step_type": "tool_result",
          "action": "received_tool_output",
          "content_preview": "Applied filter_dataframe with docstring: \n
      Filter DataFrame using a boolean condition.\n    \n    Parameters:\n
```

```
            -----------\n    df : pd.DataFrame\n        Input DataFrame\n
    condition : str\n        Boolean condition expression (e.g., 'col1 >
    0', 'col1 == \"value\"', 'col1 > 0 and col2 < 100')\n    \n...",
      "content_length": 1141
    },
    {
      "step_number": 42,
      "message_type": "HumanMessage",
      "timestamp": "2025-11-21T00:42:26.668830",
      "step_type": "reflection",
      "action": "llm_reflection",
      "content_preview": "Reasoning: The agent initially struggled with
    the correct function call syntax for loading and manipulating data,
    causing delays in early steps. It successfully loaded and
    concatenated the train and test datasets, summarized missing values,
    cast categorical columns, and separated features and target...",
      "content_length": 850,
      "extracted_score": 5.0,
      "full_reflection_content": "Reasoning: The agent initially
    struggled with the correct function call syntax for loading and
    manipulating data, causing delays in early steps. It successfully
    loaded and concatenated the train and test datasets, summarized
    missing values, cast categorical columns, and separated features and
    target variables. However, the agent overlooked checking and
    handling missing target values before model training, leading to a
    model fitting failure. Once identified, it correctly filtered out
    rows with missing targets to fix this issue. The trajectory shows
    gradual improvement and understanding but is currently incomplete as
    it has not proceeded to feature engineering, model training,
    evaluation, or prediction saving yet. The agent needs to continue
    systematically through these steps to fully solve the
    task.\n\nScore: 5\nFound_solution: False\nScore: 5"
    },
    {
      "step_number": 43,
      "message_type": "AIMessage",
      "timestamp": "2025-11-21T00:42:26.668845",
      "step_type": "tool_selection",
      "action": "selected_tools_for_execution",
      "tools_selected": [
        "convert_dataframe_to_features_target"
      ],
      "tool_calls_detail": [
        {
          "name": "convert_dataframe_to_features_target",
          "args": {
            "bindings": {
              "df": "combined_train_df"
            },
            "output": "features_target_tuple_cleaned",
            "func_kwargs": {
              "target_column": "Transported",
              "is_train": true
            }
          },
          "id": "call_rm2jT91kE90EoOciGAwUXnco",
          "type": "tool_call"
        }
      ],
      "content": ""
    },
    {
      "step_number": 44,
      "message_type": "ToolMessage",
      "timestamp": "2025-11-21T00:42:26.668849",
```

```
      "step_type": "tool_result",
      "action": "received_tool_output",
      "content_preview": "Applied convert_dataframe_to_features_target
with docstring: \n    Convert DataFrame to features and target
format.\n    \n    Parameters:\n    -----------\n    df :
pd.DataFrame\n        Input DataFrame\n    target_column : str\n
    Name of the target column\n    is_train : bool\n        Whether
the Data...",
      "content_length": 888
    },
    {
      "step_number": 45,
      "message_type": "HumanMessage",
      "timestamp": "2025-11-21T00:42:26.668854",
      "step_type": "reflection",
      "action": "llm_reflection",
      "content_preview": "Reasoning: The trajectory initially suffered
from repeated syntax errors in function calls, blocking early
progress on data loading. After correcting these, the agent
effectively loaded train and test data, summarized missing values,
concatenated datasets, and cast categorical columns properly. A
ke...",
      "content_length": 949,
      "extracted_score": 6.0,
      "full_reflection_content": "Reasoning: The trajectory initially
suffered from repeated syntax errors in function calls, blocking
early progress on data loading. After correcting these, the agent
effectively loaded train and test data, summarized missing values,
concatenated datasets, and cast categorical columns properly. A key
oversight was the presence of missing target values, which caused
model tuning to fail. The agent correctly identified this and
filtered out rows with missing targets before proceeding. The agent
has now prepared clean training data and is ready for model tuning.
However, feature engineering and model evaluation steps are still
pending. The trajectory shows a solid recovery from early mistakes
but remains incomplete in solving the full task of generating a
trained model and valid predictions. Continued systematic progress
through cleaning, feature engineering, modeling, and prediction
saving is needed to fully solve the challenge.\nScore: 6"
    },
    {
      "step_number": 46,
      "message_type": "AIMessage",
      "timestamp": "2025-11-21T00:42:26.668870",
      "step_type": "tool_selection",
      "action": "selected_tools_for_execution",
      "tools_selected": [
        "tune_lightgbm_classifier"
      ],
      "tool_calls_detail": [
        {
          "name": "tune_lightgbm_classifier",
          "args": {
            "bindings": {
              "X_train": "X_train",
              "y_train": "Y_train"
            },
            "output": "best_lgbm_model"
          },
          "id": "call_SraYlT5IClOasjkTW9oju8xt",
          "type": "tool_call"
        }
      ],
      "content": ""
    },
```

```
{
  "step_number": 47,
  "message_type": "ToolMessage",
  "timestamp": "2025-11-21T00:42:26.668873",
  "step_type": "tool_result",
  "action": "received_tool_output",
  "content_preview": "Applied tune_lightgbm_classifier with
docstring: \n    Perform hyperparameter tuning for LightGBM
Classifier using GridSearchCV.\n\n    Parameters\n    ----------\n
 X_train : pd.DataFrame\n        Training features\n    y_train :
pd.Series\n        Training target\n    cv : int, default=5\n
Number o...",
  "content_length": 1092
},
{
  "step_number": 48,
  "message_type": "HumanMessage",
  "timestamp": "2025-11-21T00:42:26.668878",
  "step_type": "reflection",
  "action": "llm_reflection",
  "content_preview": "Reasoning: The agent initially struggled with
the correct syntax for function calls, which delayed progress in
loading and concatenating data. However, the agent quickly recovered
by successfully loading both train and test datasets, summarizing
missing values, concatenating datasets, and correctly ...",
  "content_length": 1163,
  "extracted_score": 6.0,
  "full_reflection_content": "Reasoning: The agent initially
struggled with the correct syntax for function calls, which delayed
progress in loading and concatenating data. However, the agent
quickly recovered by successfully loading both train and test
datasets, summarizing missing values, concatenating datasets, and
correctly casting categorical columns. A notable oversight was the
failure to check for missing target values before model tuning,
which resulted in an error. The agent fixed this by filtering out
rows with missing targets. Subsequently, the agent performed
hyperparameter tuning for a LightGBM classifier and obtained a best
CV score of about 0.795, which is a good baseline for the
classification task. The agent has established a good foundation but
has not clearly shown subsequent steps such as evaluation on a
validation set, feature engineering enhancements, saving the best
model, or generating and saving predictions for the test set
submission file. To fully solve the challenge, these remaining steps
are needed. Overall, the trajectory shows reasonable understanding
and recovery from initial mistakes but remains incomplete in
finalizing the solution.\nScore: 6"
},
```

### I.3.1 SUCCESSFUL TRAJECTORY

This successful trajectory comprises 27 tool calls and attains a leaderboard percentile of 58.21 on our test split. It is worth noting that LATS exclusively employed the LightGBM classifier and did not consider other modeling alternatives.

```
{
  "step_number": 61,
  "message_type": "AIMessage",
  "timestamp": "2025-11-20T23:52:58.042763",
  "step_type": "tool_selection",
  "action": "selected_tools_for_execution",
  "tools_selected": [
    "convert_dataframe_to_features_target"
  ],
  "tool_calls_detail": [
```

```
        {
          "name": "convert_dataframe_to_features_target",
          "args": {
            "bindings": {
              "df": "test_df"
            },
            "func_kwargs": {
              "target_column": "Transported",
              "is_train": false
            },
            "output": "X_test"
          },
          "id": "call_LNz1Q2BWh1hYxqkPIbk29ojD",
          "type": "tool_call"
        }
      ],
      "content": ""
    },
    {
      "step_number": 62,
      "message_type": "ToolMessage",
      "timestamp": "2025-11-20T23:52:58.042766",
      "step_type": "tool_result",
      "action": "received_tool_output",
      "content_preview": "Applied convert_dataframe_to_features_target
    with docstring: \n    Convert DataFrame to features and target
    format.\n    \n    Parameters:\n    -----------\n    df :
    pd.DataFrame\n        Input DataFrame\n    target_column : str\n
      Name of the target column\n    is_train : bool\n        Whether
    the Data...",
      "content_length": 863
    },
    {
      "step_number": 63,
      "message_type": "HumanMessage",
      "timestamp": "2025-11-20T23:52:58.042770",
      "step_type": "reflection",
      "action": "llm_reflection",
      "content_preview": "Reasoning: The agent followed a generally
    appropriate and methodical trajectory, starting with loading and
    concatenating train and test datasets to allow consistent
    preprocessing. They identified and handled missing values using
    sensible imputation methods for categorical and numerical features.
    The...",
      "content_length": 1404,
      "extracted_score": 6.0,
      "full_reflection_content": "Reasoning: The agent followed a
    generally appropriate and methodical trajectory, starting with
    loading and concatenating train and test datasets to allow
    consistent preprocessing. They identified and handled missing values
    using sensible imputation methods for categorical and numerical
    features. The agent cast categorical columns correctly and performed
    meaningful feature engineering, including extracting cabin features,
    aggregating spending features, and computing group features from
    PassengerId, demonstrating good domain understanding.\n\nHowever,
    the agent made some critical mistakes in the execution, such as not
    splitting the combined dataset back into train and test before model
    training, causing NaNs in the target labels and failing model tuning
    initially. They corrected this splitting issue later but also had
    minor syntax errors in tool usage. The agent has yet to demonstrate
    the crucial steps of fitting models, hyperparameter tuning,
    evaluation, saving the best model, and generating and saving
    predictions in the required submission format and
    location.\n\nOverall, the trajectory is solid for data loading,
    cleaning, and feature engineering but incomplete for the full
```

```
challenge solution. The agent needs to proceed with modeling and
output saving to fully solve the task. The response quality is good
for preliminary steps but insufficient for a complete
solution.\n\nScore: 6\nScore: 6"
},
{
  "step_number": 64,
  "message_type": "AIMessage",
  "timestamp": "2025-11-20T23:52:58.042793",
  "step_type": "tool_selection",
  "action": "selected_tools_for_execution",
  "tools_selected": [
    "tune_lightgbm_classifier"
  ],
  "tool_calls_detail": [
    {
      "name": "tune_lightgbm_classifier",
      "args": {
        "bindings": {
          "X_train": "X_train",
          "y_train": "Y_train"
        },
        "output": "best_lgbm_model"
      },
      "id": "call_WZihuqBOZNl9QcPJNoH1PhVo",
      "type": "tool_call"
    }
  ],
  "content": ""
},
{
  "step_number": 65,
  "message_type": "ToolMessage",
  "timestamp": "2025-11-20T23:52:58.042796",
  "step_type": "tool_result",
  "action": "received_tool_output",
  "content_preview": "Applied tune_lightgbm_classifier with
docstring: \n    Perform hyperparameter tuning for LightGBM
Classifier using GridSearchCV.\n\n    Parameters\n    ----------\n
 X_train : pd.DataFrame\n        Training features\n    y_train :
pd.Series\n        Training target\n    cv : int, default=5\n
Number o...",
  "content_length": 1092
},
{
  "step_number": 66,
  "message_type": "HumanMessage",
  "timestamp": "2025-11-20T23:52:58.042802",
  "step_type": "reflection",
  "action": "llm_reflection",
  "content_preview": "Reasoning: The agent has followed a generally
sound and methodical approach suitable for the Spaceship Titanic
classification task. They correctly loaded and concatenated the
datasets, handled missing values with sensible imputation, cast
categorical variables, and performed meaningful feature engin...",
  "content_length": 1162,
  "extracted_score": 7.0,
  "full_reflection_content": "Reasoning: The agent has followed a
generally sound and methodical approach suitable for the Spaceship
Titanic classification task. They correctly loaded and concatenated
the datasets, handled missing values with sensible imputation, cast
categorical variables, and performed meaningful feature engineering
such as extracting cabin features and group-related variables. The
agent made some mistakes in tool usage and workflow order, notably
not splitting combined data before modeling, causing initial tuning
```

errors due to NaNs in the target. These were corrected later. The
agent successfully performed hyperparameter tuning on a LightGBM
classifier and obtained a reasonable CV score (~0.79), indicating a
decent model fit. However, the agent has not yet shown the steps to
save the best model or generate and save predictions on the test set
as required. The trajectory is strong in data preparation and
modeling but incomplete on final output generation and saving, which
is essential for solving the task fully. Overall, the response is
effective but partial, scoring high on foundational steps but
lacking completion of the full solution pipeline.\nScore: 7"
    },
    {
      "step_number": 67,
      "message_type": "AIMessage",
      "timestamp": "2025-11-20T23:52:58.042822",
      "step_type": "tool_selection",
      "action": "selected_tools_for_execution",
      "tools_selected": [
        "evaluate_classification_model"
      ],
      "tool_calls_detail": [
        {
          "name": "evaluate_classification_model",
          "args": {
            "bindings": {
              "model": "best_estimator",
              "X_test": "X_train",
              "y_test": "Y_train"
            },
            "func_kwargs": {
              "model_name": "LightGBM",
              "eval_data_label": "train"
            }
          },
          "id": "call_18GzPmWkNs6zNLuY9s0774Cm",
          "type": "tool_call"
        }
      ],
      "content": ""
    },
    {
      "step_number": 68,
      "message_type": "ToolMessage",
      "timestamp": "2025-11-20T23:52:58.042825",
      "step_type": "tool_result",
      "action": "received_tool_output",
      "content_preview": "Applied evaluate_classification_model with
docstring: \n    Evaluate a trained classification model on
data.\n\n    This function is used to evaluate the performance of a
trained classification model on a given dataset.\n\n    The dataset
can be the anything, such as the test set or the train set.\n\n
U...",
      "content_length": 1820
    },
    {
      "step_number": 69,
      "message_type": "HumanMessage",
      "timestamp": "2025-11-20T23:52:58.042829",
      "step_type": "reflection",
      "action": "llm_reflection",
      "content_preview": "Reasoning: The agent followed a standard and
appropriate pipeline for the Spaceship Titanic classification task
by loading and concatenating train and test datasets, imputing
missing values, casting categorical variables, and executing
meaningful feature engineering (extracting deck and side from Ca...",

```
      "content_length": 1303,
      "extracted_score": 7.0,
      "full_reflection_content": "Reasoning: The agent followed a
standard and appropriate pipeline for the Spaceship Titanic
classification task by loading and concatenating train and test
datasets, imputing missing values, casting categorical variables,
and executing meaningful feature engineering (extracting deck and
side from Cabin, aggregating spending, and creating group features).
They correctly split combined data back to train and test before
modeling, which fixed an earlier critical workflow mistake. The
agent performed hyperparameter tuning on a LightGBM classifier and
achieved a reasonable CV score (˜0.79) and training accuracy (˜81%),
indicating a decent model fit.\n\nHowever, despite the solid data
preparation and modeling, the agent did not proceed to generate
predictions on the test set or save the best model and predictions
to the required directories, which is mandatory to fully solve the
task. There were also minor syntax errors and API usage issues that
caused some delays. Overall, the trajectory is methodical and
technically sound in data handling, feature engineering, and
modeling, but incomplete as it lacks the final step of output
generation. The agent should now predict on the test set, save the
submission file as specified, and also save the best model to
conclude the task.\n\nScore: 7\nScore: 7"
    },
    {
      "step_number": 70,
      "message_type": "AIMessage",
      "timestamp": "2025-11-20T23:52:58.042851",
      "step_type": "tool_selection",
      "action": "selected_tools_for_execution",
      "tools_selected": [
        "predict_target"
      ],
      "tool_calls_detail": [
        {
          "name": "predict_target",
          "args": {
            "bindings": {
              "model": "best_estimator",
              "X_data": "X_test"
            },
            "func_kwargs": {
              "model_name": "LightGBM",
              "return_probabilities": false
            },
            "output": "test_predictions"
          },
          "id": "call_9j7gLB7G1cH6WGnnRpygvWVb",
          "type": "tool_call"
        }
      ],
      "content": ""
    },
    {
      "step_number": 71,
      "message_type": "ToolMessage",
      "timestamp": "2025-11-20T23:52:58.042854",
      "step_type": "tool_result",
      "action": "received_tool_output",
      "content_preview": "Applied predict_target with docstring: \n
Make predictions using a trained model.\n    \n    Parameters\n
----------\n    model : Any\n        Trained model (regression or
classification)\n    X_data : pd.DataFrame\n        Features to make
predictions on\n    model_name : str, default=\"model\"\n
...",
```

```json
        "content_length": 1018
    },
    {
      "step_number": 72,
      "message_type": "HumanMessage",
      "timestamp": "2025-11-20T23:52:58.042858",
      "step_type": "reflection",
      "action": "llm_reflection",
      "content_preview": "Reasoning: The agent followed a comprehensive
and well-structured approach to the Spaceship Titanic classification
task. They started by loading and concatenating the train and test
datasets to ensure consistent preprocessing. They successfully
identified missing data and applied appropriate imputat...",
      "content_length": 1588,
      "extracted_score": 7.0,
      "full_reflection_content": "Reasoning: The agent followed a
comprehensive and well-structured approach to the Spaceship Titanic
classification task. They started by loading and concatenating the
train and test datasets to ensure consistent preprocessing. They
successfully identified missing data and applied appropriate
imputation strategies for categorical and numerical features.
Categorical columns were correctly cast, and meaningful feature
engineering was performed, including extracting cabin deck and side,
aggregating spending features, and creating group features from
PassengerId, demonstrating good domain understanding. The agent then
corrected a critical oversight by properly splitting the combined
dataset back into train and test before modeling, which allowed
successful hyperparameter tuning of a LightGBM model, yielding a
reasonable cross-validation score (~0.79) and training accuracy
(~81%). Evaluation results were clearly reported. Finally, the agent
generated predictions on the test data. However, the agent did not
proceed to save the best model or the predictions to the specified
submission folder, which is essential to fully solve the task as per
instructions. There were also some minor API usage errors and
workflow missteps early on, but these were quickly corrected.
Overall, the agent demonstrated good understanding and execution of
data preparation, feature engineering, and modeling. The main
shortcoming is not completing the final output saving step, leaving
the task only partially solved. The trajectory is methodical, mostly
sufficient, but incomplete. Score: 7\nScore: 7"
    },
    {
      "step_number": 73,
      "message_type": "AIMessage",
      "timestamp": "2025-11-20T23:52:58.042883",
      "step_type": "tool_selection",
      "action": "selected_tools_for_execution",
      "tools_selected": [
        "save_dataframe_to_csv"
      ],
      "tool_calls_detail": [
        {
          "name": "save_dataframe_to_csv",
          "args": {
            "df": "test_predictions",
            "filepath":
"submissions/spaceship_titanic/lats_reflection/lats_reflection.csv"
          },
          "id": "call_BTgHyO2TJNUOWcyx4YTiNwIr",
          "type": "tool_call"
        }
      ],
      "content": ""
    },
```

## I.4 MCTS-Outcome

### I.4.1 Failed Trajectory

The most promising trajectory identified consisted of 18 tool calls but repeatedly encountered failures when attempting to load the data. Consequently, the entire search tree remained unproductive and failed to reach the modeling phase. This suggests that the system lacked sufficient guidance to advance beyond the initial stages of the pipeline.

```
{
  "step_number": 22,
  "message_type": "AIMessage",
  "timestamp": "2025-11-20T16:12:41.763597",
  "step_type": "tool_selection",
  "action": "selected_tools_for_execution",
  "tools_selected": [
    "read_data"
  ],
  "tool_calls_detail": [
    {
      "name": "read_data",
      "args": {
        "output": "train_df"
      },
      "id": "call_3did4EVkLGDyw9QRFoiAnUlA",
      "type": "tool_call"
    }
  ],
  "content": ""
},
{
  "step_number": 23,
  "message_type": "ToolMessage",
  "timestamp": "2025-11-20T16:12:41.763599",
  "step_type": "tool_result",
  "action": "received_tool_output",
  "content_preview": "Error: TypeError(\"read_data() missing 1
required positional argument: 'filepath'\")\n Please fix your
mistakes.",
  "content_length": 109
},
```

### I.4.2 Successful Trajectory

This trajectory contains 32 tool calls and fits only a CatBoost model. Notably, the submission file produced by this run exhibited a shape mismatch with the hidden test set, resulting in a percentile score of 0. The mismatch arose from errors in the feature engineering stage, which filtered out a portion of the test data.

```
{
  "step_number": 63,
  "message_type": "ToolMessage",
  "timestamp": "2025-11-20T16:13:57.197079",
  "step_type": "tool_result",
  "action": "received_tool_output",
  "content_preview": "Applied label_encode with docstring: \n
Label encode categorical columns.\n    \n    Parameters:\n
----------\n    df : pd.DataFrame\n        Input DataFrame\n
columns : str or List[str], optional\n        Column(s) to encode.
If None, encodes all object/category columns.\n    \n    Returns:\n
---...",
  "content_length": 701
},
{
```

```
      "step_number": 64,
      "message_type": "HumanMessage",
      "timestamp": "2025-11-20T16:13:57.197084",
      "step_type": "reward_feedback",
      "action": "generated_reward_feedback",
      "content_preview": "Modeling is still in progress",
      "content_length": 29
    },
    {
      "step_number": 65,
      "message_type": "AIMessage",
      "timestamp": "2025-11-20T16:13:57.197089",
      "step_type": "tool_selection",
      "action": "selected_tools_for_execution",
      "tools_selected": [
        "tune_catboost_classifier"
      ],
      "tool_calls_detail": [
        {
          "name": "tune_catboost_classifier",
          "args": {
            "bindings": {
              "X_train": "X_train",
              "y_train": "Y_train"
            },
            "output": "best_catboost_model"
          },
          "id": "call_WbNtNSt3M3UXFDk5JBj49Nkm",
          "type": "tool_call"
        }
      ],
      "content": ""
    },
    {
      "step_number": 66,
      "message_type": "ToolMessage",
      "timestamp": "2025-11-20T16:13:57.197092",
      "step_type": "tool_result",
      "action": "received_tool_output",
      "content_preview": "Applied tune_catboost_classifier with
docstring: \n    Perform hyperparameter tuning for CatBoost
Classifier using GridSearchCV.\n\n    Parameters\n    ----------\n
 X_train : pd.DataFrame\n        Training features\n    y_train :
pd.Series\n        Training target\n    cv : int, default=5\n
Number o...",
      "content_length": 1087
    },
    {
      "step_number": 67,
      "message_type": "HumanMessage",
      "timestamp": "2025-11-20T16:13:57.197097",
      "step_type": "reward_feedback",
      "action": "generated_reward_feedback",
      "content_preview": "Human Feedback: Verified that the modeling was
successful",
      "content_length": 57
    },
    {
      "step_number": 68,
      "message_type": "AIMessage",
      "timestamp": "2025-11-20T16:13:57.197102",
      "step_type": "tool_selection",
      "action": "selected_tools_for_execution",
      "tools_selected": [
        "evaluate_classification_model"
```

```
          ],
          "tool_calls_detail": [
            {
              "name": "evaluate_classification_model",
              "args": {
                "bindings": {
                  "model": "best_estimator",
                  "X_test": "X_train",
                  "y_test": "Y_train"
                },
                "func_kwargs": {
                  "model_name": "CatBoostClassifier",
                  "eval_data_label": "train"
                }
              },
              "id": "call_hKlPSrF4ZrPiof5KJTUGpsOb",
              "type": "tool_call"
            }
          ],
          "content": ""
        },
        {
          "step_number": 69,
          "message_type": "ToolMessage",
          "timestamp": "2025-11-20T16:13:57.197105",
          "step_type": "tool_result",
          "action": "received_tool_output",
          "content_preview": "Applied evaluate_classification_model with
      docstring: \n    Evaluate a trained classification model on
      data.\n\n    This function is used to evaluate the performance of a
      trained classification model on a given dataset.\n\n    The dataset
      can be the anything, such as the test set or the train set.\n\n
      U...",
          "content_length": 1830
        },
        {
          "step_number": 70,
          "message_type": "HumanMessage",
          "timestamp": "2025-11-20T16:13:57.197112",
          "step_type": "reward_feedback",
          "action": "generated_reward_feedback",
          "content_preview": "Human Feedback: submission DataFrame was NOT
      created successfully. Please check the signature of the wrapped
      function if any, and call it with the correct arguments. The tool
      signature is This tool reads arguments from the scratchpad using
      `bindings`, passes them to the internal function.\n\n          A...",
          "content_length": 1979
        },
        {
```

## I.5 MCTS-Shaped

### I.5.1 Failed Trajectory

The most promising trajectory consisted of 19 steps and terminated at the stage of splitting the combined data into train and test. This indicates that the search never reached the modeling phase.

```
          {
          "step_number": 49,
          "message_type": "AIMessage",
          "timestamp": "2025-11-20T17:41:18.892989",
          "step_type": "tool_selection",
          "action": "selected_tools_for_execution",
          "tools_selected": [
```

```
            "drop_feature"
          ],
          "tool_calls_detail": [
            {
              "name": "drop_feature",
              "args": {
                "bindings": {
                  "df": "combined_df"
                },
                "output": "combined_df",
                "func_kwargs": {
                  "column": [
                    "PassengerId",
                    "Name",
                    "Cabin"
                  ]
                }
              },
              "id": "call_W8BQW8V16Jch32pPwgSkJXIN",
              "type": "tool_call"
            }
          ],
          "content": ""
        },
        {
          "step_number": 50,
          "message_type": "ToolMessage",
          "timestamp": "2025-11-20T17:41:18.892992",
          "step_type": "tool_result",
          "action": "received_tool_output",
          "content_preview": "Applied drop_feature with docstring: \n
Drop feature(s) from the DataFrame.\n    \n    Parameters:\n
----------\n    df : pd.DataFrame\n        Input DataFrame\n
column : str or List[str]\n        Name(s) of the column(s) to
drop\n    \n    Returns:\n    --------\n    pd.DataFrame\n
DataFrame wi...",
          "content_length": 662
        },
        {
          "step_number": 51,
          "message_type": "HumanMessage",
          "timestamp": "2025-11-20T17:41:18.892997",
          "step_type": "reward_feedback",
          "action": "generated_reward_feedback",
          "content_preview": "Categorical columns found: [{'column':
'HomePlanet', 'unique_count': 3, 'dtype': 'category'}, {'column':
'CryoSleep', 'unique_count': 3, 'dtype': 'category'}, {'column':
'Destination', 'unique_count': 4, 'dtype': 'category'}]. Columns
with dtypes that are not numeric/categorical found: []. Please
co...",
          "content_length": 525
        },
        {
          "step_number": 52,
          "message_type": "AIMessage",
          "timestamp": "2025-11-20T17:41:18.893002",
          "step_type": "tool_selection",
          "action": "selected_tools_for_execution",
          "tools_selected": [
            "label_encode"
          ],
          "tool_calls_detail": [
            {
              "name": "label_encode",
              "args": {
```

```
            "bindings": {
              "df": "combined_df"
            }
          },
          "id": "call_LZuEYLrfsTWPCfWsh5Ungldl",
          "type": "tool_call"
        }
      ],
      "content": ""
    },
    {
      "step_number": 53,
      "message_type": "ToolMessage",
      "timestamp": "2025-11-20T17:41:18.893005",
      "step_type": "tool_result",
      "action": "received_tool_output",
      "content_preview": "Applied label_encode with docstring: \n
Label encode categorical columns.\n    \n    Parameters:\n
-----------\n    df : pd.DataFrame\n        Input DataFrame\n
columns : str or List[str], optional\n        Column(s) to encode.
If None, encodes all object/category columns.\n    \n    Returns:\n
    ---...",
      "content_length": 709
    },
    {
      "step_number": 54,
      "message_type": "HumanMessage",
      "timestamp": "2025-11-20T17:41:18.893010",
      "step_type": "reward_feedback",
      "action": "generated_reward_feedback",
      "content_preview": "Human Feedback: Verified that the feature
engineering is complete and all columns are numeric (int/float)",
      "content_length": 105
    },
    {
      "step_number": 55,
      "message_type": "AIMessage",
      "timestamp": "2025-11-20T17:41:18.893015",
      "step_type": "tool_selection",
      "action": "selected_tools_for_execution",
      "tools_selected": [
        "split_combined_into_train_test"
      ],
      "tool_calls_detail": [
        {
          "name": "split_combined_into_train_test",
          "args": {
            "bindings": {
              "combined": "combined_df"
            },
            "output": "splitted_dfs"
          },
          "id": "call_LsWLDkHhkrEQ8Ja0rrBMQOXh",
          "type": "tool_call"
        }
      ],
      "content": ""
    },
    {
      "step_number": 56,
      "message_type": "ToolMessage",
      "timestamp": "2025-11-20T17:41:18.893018",
      "step_type": "tool_result",
      "action": "received_tool_output",
```

```
      "content_preview": "Applied split_combined_into_train_test with
    docstring: \n    Split combined data back into train and test using
    tracking columns.\n    \n    Parameters:\n    -----------\n
    combined : pd.DataFrame\n        Combined DataFrame with tracking
    columns\n    \n    Returns:\n    --------\n    tuple\n
    (train_d...",
      "content_length": 607
    },
    {
      "step_number": 57,
      "message_type": "HumanMessage",
      "timestamp": "2025-11-20T17:41:18.893022",
      "step_type": "reward_feedback",
      "action": "generated_reward_feedback",
      "content_preview": "Human Feedback: Verified that the train and
    test data were split successfully",
      "content_length": 77
    }
  ],
```

### I.5.2  SUCCESSFUL TRAJECTORY

The trajectory spans 23 steps and fits exclusively a LightGBM classifier in the modeling stage. When evaluated on our test split, it attains a leaderboard percentile of 33.11.

```
    {
      "step_number": 61,
      "message_type": "AIMessage",
      "timestamp": "2025-11-20T17:45:34.047940",
      "step_type": "tool_selection",
      "action": "selected_tools_for_execution",
      "tools_selected": [
        "tune_lightgbm_classifier"
      ],
      "tool_calls_detail": [
        {
          "name": "tune_lightgbm_classifier",
          "args": {
            "bindings": {
              "X_train": "X_train",
              "y_train": "Y_train"
            },
            "output": "tuned_model_info"
          },
          "id": "call_LfolkEfImAXxyb5Ha3nNW4cp",
          "type": "tool_call"
        }
      ],
      "content": ""
    },
    {
      "step_number": 62,
      "message_type": "ToolMessage",
      "timestamp": "2025-11-20T17:45:34.047944",
      "step_type": "tool_result",
      "action": "received_tool_output",
      "content_preview": "Applied tune_lightgbm_classifier with
    docstring: \n    Perform hyperparameter tuning for LightGBM
    Classifier using GridSearchCV.\n\n    Parameters\n    ----------\n
     X_train : pd.DataFrame\n        Training features\n    y_train :
    pd.Series\n        Training target\n    cv : int, default=5\n
    Number o...",
      "content_length": 1092
    },
```

```
      {
        "step_number": 63,
        "message_type": "HumanMessage",
        "timestamp": "2025-11-20T17:45:34.047949",
        "step_type": "reward_feedback",
        "action": "generated_reward_feedback",
        "content_preview": "Human Feedback: Verified that the modeling was
    successful",
        "content_length": 57
      },
      {
        "step_number": 64,
        "message_type": "AIMessage",
        "timestamp": "2025-11-20T17:45:34.047955",
        "step_type": "tool_selection",
        "action": "selected_tools_for_execution",
        "tools_selected": [
          "evaluate_classification_model"
        ],
        "tool_calls_detail": [
          {
            "name": "evaluate_classification_model",
            "args": {
              "bindings": {
                "model": "best_estimator",
                "X_test": "X_train",
                "y_test": "Y_train"
              },
              "func_kwargs": {
                "model_name": "LightGBM",
                "eval_data_label": "train"
              }
            },
            "id": "call_iLtqzfkOBwtWmROl9WDHKO75",
            "type": "tool_call"
          }
        ],
        "content": ""
      },
      {
        "step_number": 65,
        "message_type": "ToolMessage",
        "timestamp": "2025-11-20T17:45:34.047959",
        "step_type": "tool_result",
        "action": "received_tool_output",
        "content_preview": "Applied evaluate_classification_model with
    docstring: \n    Evaluate a trained classification model on
    data.\n\n    This function is used to evaluate the performance of a
    trained classification model on a given dataset.\n\n    The dataset
    can be the anything, such as the test set or the train set.\n\n
    U...",
        "content_length": 1820
      },
      {
        "step_number": 66,
        "message_type": "HumanMessage",
        "timestamp": "2025-11-20T17:45:34.047965",
        "step_type": "reward_feedback",
        "action": "generated_reward_feedback",
        "content_preview": "Human Feedback: submission DataFrame was NOT
    created successfully. Please check the signature of the wrapped
    function if any, and call it with the correct arguments. The tool
    signature is This tool reads arguments from the scratchpad using
    `bindings`, passes them to the internal function.\n\n         A...",
        "content_length": 1979
```

```
      },
      {
        "step_number": 67,
        "message_type": "AIMessage",
        "timestamp": "2025-11-20T17:45:34.047972",
        "step_type": "tool_selection",
        "action": "selected_tools_for_execution",
        "tools_selected": [
          "predict_target"
        ],
        "tool_calls_detail": [
          {
            "name": "predict_target",
            "args": {
              "bindings": {
                "model": "best_estimator",
                "X_data": "X_test"
              },
              "output": "test_predictions",
              "func_kwargs": {
                "model_name": "LightGBM"
              }
            },
            "id": "call_lZQQw4OoT0LANXYCUvjXh1Bp",
            "type": "tool_call"
          }
        ],
        "content": ""
      },
      {
        "step_number": 68,
        "message_type": "ToolMessage",
        "timestamp": "2025-11-20T17:45:34.047975",
        "step_type": "tool_result",
        "action": "received_tool_output",
        "content_preview": "Applied predict_target with docstring: \n
    Make predictions using a trained model.\n    \n    Parameters\n
    ----------\n    model : Any\n        Trained model (regression or
    classification)\n    X_data : pd.DataFrame\n        Features to make
    predictions on\n    model_name : str, default=\"model\"\n
    ...",
        "content_length": 1018
      },
      {
        "step_number": 69,
        "message_type": "HumanMessage",
        "timestamp": "2025-11-20T17:45:34.047981",
        "step_type": "reward_feedback",
        "action": "generated_reward_feedback",
        "content_preview": "Human Feedback: Verified that the submission
    DataFrame was created successfully with 1 columns (boolean) and no
    missing values.",
        "content_length": 127
      }
    ],
```

## I.6 HIERARCHICAL MCTS

### I.6.1 FAILED TRAJECTORY

This trajectory consists of 55 tool calls and fails at the train data to features and target stage, prior to modeling. Nothing of note in this trajectory, and hence is not reported.

### I.6.2 SUCCESSFUL TRAJECTORY

The trajectory spans 80 steps and evaluates multiple modeling approaches, attaining a leaderboard percentile of 62.65. Hierarchical MCTS demonstrates the ability to execute substantially longer trajectories, owing to its use of subtask decomposition and task masking.

```json
{
        "step_type": "tool_selection",
        "action": "selected_tools_for_execution",
        "tools_selected": [
          "tune_lightgbm_classifier"
        ],
        "tool_calls_detail": [
          {
            "name": "tune_lightgbm_classifier",
            "args": {
              "bindings": {
                "X_train": "X_train",
                "y_train": "Y_train"
              },
              "output": "tuned_lgbm_classifier"
            },
            "id": "call_fM1ezojD8q9aj9f052GtppXm",
            "type": "tool_call"
          }
        ],
        "content": ""
      },
      {
        "step_type": "tool_result",
        "action": "received_tool_output",
        "content_preview": "Applied tune_lightgbm_classifier with
    docstring: \n    Perform hyperparameter tuning for LightGBM
    Classifier using GridSearchCV.\n\n    Parameters\n    ----------\n
     X_train : pd.DataFrame\n        Training features\n    y_train :
    pd.Series\n        Training target\n    cv : int, default=5\n
    Number o...",
        "content_length": 1092
      },
      {
        "step_type": "reward_feedback",
        "action": "generated_reward_feedback",
        "content_preview": "Human Feedback: Verified that the modeling
    was successful",
        "content_length": 57
      },
{
        "step_type": "tool_selection",
        "action": "selected_tools_for_execution",
        "tools_selected": [
          "tune_random_forest_classifier"
        ],
        "tool_calls_detail": [
          {
            "name": "tune_random_forest_classifier",
            "args": {
              "bindings": {
                "X_train": "X_train",
                "y_train": "Y_train"
              },
              "func_kwargs": {
                "cv": 5
              },
              "output": "rf_tuned_model_output"
            },
```

```
                "id": "call_3OsXBYeYfbCnehW6uvlfA99X",
                "type": "tool_call"
              }
            ],
            "content": "I cannot evaluate the model on the test data
      because the true target labels are not available for the test set.
      Instead, I will generate predictions for the test data using the
      best Random Forest classifier model and save the predictions to the
      required submission file path."
          },
          {
            "step_type": "tool_result",
            "action": "received_tool_output",
            "content_preview": "Applied tune_random_forest_classifier with
      docstring: \n    Perform hyperparameter tuning for Random Forest
      Classifier using GridSearchCV.\n\n    Parameters\n    ----------\n
       X_train : pd.DataFrame        Training features\n    y_train :
      pd.Series        Training target\n    cv : int, default=5\n
      ...",
            "content_length": 1134
          },
          {
            "step_type": "reward_feedback",
            "action": "generated_reward_feedback",
            "content_preview": "Human Feedback: Verified that the modeling
      was successful",
            "content_length": 57
```

