# OpenReview forum: "ML-Tool-Bench: Tool-Augmented Planning for ML Tasks"
_ICLR.cc/2026/Conference — Submitted to ICLR 2026_

### Official Review · Reviewer_tCJh · 2025-10-26

**Soundness:** 3
**Presentation:** 3
**Contribution:** 3
**Rating:** 4
**Confidence:** 3

**Summary:**

This paper focuses on the tool-augmented planning capabilities of autonomous machine learning (ML) agents. Addressing the limitations of existing solutions, it proposes a benchmark framework named ML-Tool-Bench and designs improved algorithms to enhance the planning effectiveness for complex ML tasks. Finally, through experiments, the value of the framework and algorithms is verified, laying a foundation for the development of tool-augmented ML agents.

**Strengths:**

1. This work effectively addresses the critical gap in existing tool-use benchmarks that overlook long-horizon planning capabilities essential for complex ML workflows, as ML-Tool-Bench, equipped with 61 task-specific tools and 15 Kaggle tabular challenges.
2. The proposed in-memory named object management (Scratchpad) represents a key technical innovation, resolving the long-standing issue of intermediate result reuse in traditional benchmarks by allowing agents to name, store, and reference complex artifacts via key-value pairs, thus preventing global data corruption from single erroneous operations and enhancing the flexibility of multi-step trajectory execution.
3. The study demonstrates strong problem-driven innovation by first diagnosing the inherent limitations of mainstream planning methods  and then designing targeted improvements—MCTS-Shaped with staged shaped rewards and textual feedback, and Hierarchical MCTS with subtask decomposition and tool masking—ensuring the proposed solutions directly tackle core pain points.
4. The experimental covers two representative LLMs (GPT-4o and GPT-4.1-mini) to verify generalizability across model capabilities, compares five algorithms (3 baselines + 2 proposed methods) using dual metrics of trajectory consistency and Kaggle leaderboard percentile, and adopts dataset sampling and internal test splits to mimic real-world constraints, with quantitative results.

**Weaknesses:**

1. I’m not entirely clear on the purpose of designing a new benchmark framework. Existing tool-planning frameworks already address many realistic and problem-specific scenarios, along with supporting tools and prior-knowledge categorizations. For instance, ToolBench already contains extensive relevant content for such purposes. Is the main goal here to adapt better to ML-specific contexts? Fundamentally, I believe datasets should still be more aligned with real-world tasks.

2. The two improved algorithms seem mainly to modify the MCTS process flow. Compared with heuristic methods such as ToolChain*, what specific advantages do these revised versions of MCTS provide?

3. If one wants to further extend the toolset or include more tools, can the dataset proposed in this paper be expanded accordingly? At present, doesn’t the variety of available tools seem somewhat limited?

4. In this study, both the “structured textual feedback” in MCTS-Shaped and the “sub-task search” in Hierarchical MCTS assume that feedback is instantaneous and computational resources are sufficient. However, two key constraints present in real-world industrial settings are not considered:

(A)When tool calls involve large-scale data processing, results may be delayed by minutes. Such latency could desynchronize MCTS trajectory updates and reward calculations, affecting planning efficiency, yet this scenario is not simulated in the experiments.

(B)MCTS-based algorithms require multiple searches and simulations, while LLM calls (especially GPT-4o) are expensive. The experiments do not report a “call-count vs. performance” trade-off (e.g., whether Hierarchical MCTS’s gain over ReAct justifies its extra computation cost) nor explore cost-optimization strategies (such as pruning ineffective search paths).

**Questions:**

See weakness.

---

> ### Author Response · Authors · 2025-11-24
> **Response to Reviewer tCJh-Part 1**
>
> We sincerely thank the reviewer for their thoughtful feedback and insightful questions. We address their concerns both in the common response section and in the specific responses below.
>
> *1. I'm not entirely clear on the purpose of designing a new benchmark framework. Existing tool-planning frameworks already address many realistic and problem-specific scenarios, along with supporting tools and prior-knowledge categorizations. For instance, ToolBench already contains extensive relevant content for such purposes. Is the main goal here to adapt better to ML-specific contexts? Fundamentally, I believe datasets should still be more aligned with real-world tasks.*
>
> ML-Tool-Bench is the first benchmark specifically designed to evaluate long-horizon planning capability of LLMs in multi-step ML workflows, requiring 20+ sequential tool invocations without explicit guidance. Unlike existing ML benchmarks that focus on code generation or tool benchmarks that evaluate tool selection/argument extraction over 1-3 steps, our benchmark assesses whether LLMs can plan complex solutions through multi-step planning, tool composition, and sequential decision-making across trajectories reaching up to 80 steps. We refer the reviewer to the common response on Novelty for additional details and a detailed comparison table.
>
> A key distinction is that ToolBench and similar benchmarks (BFCL) evaluate whether LLMs can correctly identify which tool to invoke and extract appropriate arguments given explicit context. These are primarily retrieval and parsing tasks, typically requiring 1-3 tool calls. However, ML-Tool-Bench evaluates whether LLMs can plan and execute complex, multi-step workflows requiring 20+ sequential tool invocations, with an observed maximum of 80 tool invocations,  without explicit guidance. Any approach that solves a task on our benchmark, must discover solution paths, manage intermediate artifacts, and coordinate tools across multiple stages (data loading → cleaning → feature engineering → modeling → evaluation).
>
> We argue that our benchmark is aligned with real-world tasks. Our challenges are sampled from actual Kaggle competitions that represent genuine problems that data scientists solve in industry: Churn prediction (Kaggle challenges: Binary Classification with a Bank Churn Dataset, Santander Customer Transaction Prediction), Fraud detection (Porto Seguro Safe Driver Prediction), Demand forecasting (New York City Taxi Fare Prediction, New York City Taxi Trip Duration), and Customer segmentation (Costa Rican Household Poverty Prediction). The 61-tool set mirrors standard ML libraries (scikit-learn, XGBoost, LightGBM, pandas) used in practice. While we focus on tabular tasks for tractability and cost, the planning challenges such as tool coordination, artifact management, multi-step reasoning generalize to broader ML engineering scenarios.

---

> ### Author Response · Authors · 2025-11-24
> **Response to Reviewer tCJh-Part 2**
>
> We sincerely thank the reviewer for their thoughtful feedback and insightful questions. We address their concerns both in the common response section and in the specific responses below. We continue from Part-1.
>
> *2. The two improved algorithms seem mainly to modify the MCTS process flow. Compared with heuristic methods such as ToolChain*, what specific advantages do these revised versions of MCTS provide?*
>
> The fundamental challenge in tool-augmented planning for ML workflows is accurately estimating state value. In our setting, states comprise:
>   - Conversational history: All reasoning messages, tool invocations, tool outputs, and feedback messages accumulated over 20+ steps
>   - Complex artifacts: Trained models, transformed dataframes, and intermediate results stored in the scratchpad
>
> Estimating the value of such rich, heterogeneous states is significantly harder than in text-based tool-use scenarios.
>
> ToolChain* [Zhuang et al., 2023] employs A* search with heuristic cost functions g(n) (cumulative cost) and h(n) (estimated future cost). In ToolChain*'s text-based ToolBench setting, these heuristics use measures like Longest Common Subsequence (LCS) between trajectories, which work well for comparing textual plans. However, such heuristics are ill-suited for ML workflows: Comparing dataframe transformations or model training states via LCS is not meaningful, and crafting good heuristics requires domain expertise and extensive tuning for each stage of the ML pipeline
>
> MCTS provides critical advantages for our setting:
>
>   1. Built-in exploration: The selection policy we use balances exploration and exploitation, preventing our approaches from prematurely converging to locally optimal solutions—a key risk when value estimates are biased, noisy or unreliable
>   2. No heuristic engineering required: MCTS only needs reward signals (which we can verify deterministically) rather than hand-designed cost functions
>   3. Graceful handling of uncertainty: MCTS naturally handles value uncertainty through iterative refinement via backpropagation, whereas A* with poor heuristics can commit to suboptimal paths early
>
> Our shaped rewards and hierarchical decomposition further address the value estimation challenge by providing intermediate credit and reducing the effective state space, making MCTS tractable even without learned value functions.

---

> > ### Author Response · Authors · 2025-11-24
> > **Response to Reviewer tCJh-Part 3**
> >
> > We sincerely thank the reviewer for their thoughtful feedback and insightful questions. We address their concerns both in the common response section and in the specific responses below. We continue from Part-2.
> >
> > *3. If one wants to further extend the toolset or include more tools, can the dataset proposed in this paper be expanded accordingly? At present, doesn't the variety of available tools seem somewhat limited?*
> >
> > ML-Tool-Bench is designed to be extensible: users can add custom tools using our decorator-based interface (Section E). In theory, the toolset can be expanded arbitrarily. However, empirical evidence suggests a critical trade-off: as the number of available tools increases, performance significantly deteriorates [1]. This is because:
> >   - The action space grows, making exploration less efficient
> >   - Tool selection accuracy decreases with larger candidate sets
> >   - LLMs struggle to maintain coherent plans across more complex choice landscapes
> >
> > We believe hierarchical decomposition (Section 4.5) provides a promising path for scaling to larger toolsets. By decomposing tasks into subtasks with tool masking, the effective action space at each stage remains bounded (20-30 tools rather than 100+). Our results support this: Hierarchical MCTS significantly outperforms flat approaches, suggesting that structure is essential as toolsets grow. The results in Appendix F demonstrate that the performance of Hierarchical MCTS degrades substantially without tool masking. Specifically, Hierarchical MCTS without tool masking achieves a median consistency of 0.3 and a median leaderboard percentile position of 0 across the chosen subset of five challenges. In comparison, Hierarchical MCTS with tool masking achieves a median consistency and leaderboard percentile of 0.8 and 21.10 respectively. This highlights that both tool masking and subtask decomposition are critical for effectively solving long-horizon planning problems in high-dimensional action spaces using LLMs. However, we acknowledge that empirical validation with tool sets exceeding 100+ tools is needed to fully understand scaling behavior and identify potential bottlenecks.
> >
> > The 61 tools in our benchmark are designed to be sufficient for tabular Kaggle ML challenges (regression and classification), covering common data cleaning operations (handling missing values, type casting), diverse feature engineering techniques (aggregations, encoding, datetime extraction), and tree-based models that excel on tabular data (Random Forest, XGBoost, LightGBM, CatBoost). This represents a realistic set of operations used by practitioners on Kaggle and in industry for tabular workflows. In line with the evolution of benchmarks like BFCL (which expanded from basic function calling to multi-turn scenarios), we envision future extensions: larger tool sets with aggressive hierarchical decomposition, non-tabular modalities (computer vision, NLP) requiring tools for training neural networks, and expanded challenge sets. Our current benchmark serves as a starting point for systematic evaluation of tool-augmented planning capability, and we view these extensions as valuable future work.
> >
> > [1] Invocable APIs derived from NL2SQL datasets for LLM Tool-Calling Evaluation, Elder et al., 2025.

---

> > > ### Author Response · Authors · 2025-11-24
> > > **Response to Reviewer tCJh-Part 4**
> > >
> > > We sincerely thank the reviewer for their thoughtful feedback and insightful questions. We address their concerns both in the common response section and in the specific responses below. We continue from Part-3.
> > >
> > > *4. In this study, both the "structured textual feedback" in MCTS-Shaped and the "sub-task search" in Hierarchical MCTS assume that feedback is instantaneous and computational resources are sufficient. However, two key constraints present in real-world industrial settings are not considered:*
> > >
> > >    - **(A)** When tool calls involve large-scale data processing, results may be delayed by minutes. Such latency could desynchronize MCTS trajectory updates and reward calculations, affecting planning efficiency, yet this scenario is not simulated in the experiments.
> > >
> > >   Our current implementation waits for each tool call to finish executing before resuming planning, making the setting fully synchronous. Tool execution latency does not affect MCTS trajectory updates or reward calculations in our experiments, as the agent only proceeds after receiving complete tool outputs.
> > >
> > >   We acknowledge that real-world industrial settings may involve asynchronous tool execution with delayed results. However, the primary purpose of our benchmark is to evaluate the planning capabilities in tool-augmented settings: specifically, whether an approach can discover valid multi-step solution trajectories. We deliberately chose a synchronous setting to isolate and measure planning ability without confounding factors from asynchronous coordination or timeout handling. Extending to asynchronous scenarios with latency simulation is valuable future work, but orthogonal to our current focus on understanding fundamental planning limitations.
> > >
> > > - **(B)** MCTS-based algorithms require multiple searches and simulations, while LLM calls (especially GPT-4o) are expensive. The experiments do not report a "call-count vs. performance" trade-off (e.g., whether Hierarchical MCTS's gain over ReAct justifies its extra computation cost) nor explore cost-optimization strategies (such as pruning ineffective search paths).
> > >
> > > We refer the reviewer to the common response section on Benchmark Size for detailed cost analysis. In summary, our evaluation suite cost $5000 total (10 trajectories × 5 approaches × 15 challenges × 2 models), and LATS is 3.5x more expensive than our successful variants (Hierarchical-MCTS and MCTS-Shaped) while achieving only 0.2 consistency and 0 median percentile, demonstrating that Hierarchical MCTS's gains strongly justify its computational cost compared to both expensive methods (LATS) and cheap but ineffective ones (ReAct).

---

> > > > ### Comment · Reviewer_tCJh · 2025-11-25
> > > > **Thanks for your response**
> > > >
> > > > Thanks for your response, However, the purpose of "ML-Tool-Bench is the first benchmark specifically designed to evaluate long-horizon planning capability of LLMs in multi-step ML workflows, requiring 20+ sequential tool invocations without explicit guidance." is not that clear on your paper, also existing benchmark has the ability to evaluate whether a purpose requires short or long tool invocation. Also the two MCTS algorithms are more like a industrial revision compared to the traditional MCTS/LATS/A*. Therefore, I would like to maintain my score, but your rebuttal do strengthen the soundness of paper.

---

> > > > > ### Author Response · Authors · 2025-11-26
> > > > > **Response to Reviewer tCJh24**
> > > > >
> > > > > We appreciate the reviewer for engaging thoughtfully with our rebuttal and providing valuable feedback.
> > > > >
> > > > > We agree the paper's core contribution could be clearer. We will revise to:
> > > > >
> > > > > 1. **Lead with the benchmark's unique focus**: After extensive review of related work, to the best of our knowledge, ML-Tool-Bench is the first benchmark for evaluating *long-horizon planning with complex artifact management* (20-80 tool calls managing dataframes/models), distinct from existing tool benchmarks that evaluate *tool selection and argument extraction* (1-3 calls with primitive types).
> > > > >
> > > > > 2. **Clarify what existing benchmarks don't evaluate**: We'll make explicit upfront that ToolBench and similar benchmarks assess whether LLMs can identify the correct tool and extract arguments, primarily retrieval tasks, while our benchmark evaluates whether LLMs can discover multi-step solution paths requiring state management across 20+ sequential decisions or tool invocations.
> > > > >
> > > > > 3. **Emphasize novel components**: While MCTS itself is not new, our *deterministic shaped rewards with stage-based verification* represent a novel contribution that addresses a critical failure mode: LLM-based evaluation (used in LATS) produces inconsistent scoring for long trajectories, achieving only 0.2 consistency while costing 3.5x more than our approach. Our deterministic verification provides objective, reproducible rewards essential for long-horizon planning. Additionally, *hierarchical decomposition with tool masking* is a novel methodology for managing large action spaces in tool-augmented settings.
> > > > >
> > > > > We're grateful for the reviewer's feedback and will incorporate these clarifications in the revised paper.

---

### Official Review · Reviewer_f1i6 · 2025-10-26

**Soundness:** 2
**Presentation:** 2
**Contribution:** 2
**Rating:** 4
**Confidence:** 4

**Summary:**

This paper presents ML-Tool-Bench, a benchmark and framework designed to evaluate large language models on ML-related tool-augmented planning and execution tasks. The authors argue that prior benchmarks (e.g., MLE-Bench, MLAgentBench) mainly test end-to-end ML problem solving but overlook the crucial intermediate skill of decomposing ML tasks into tool calls and reasoning sequences. ML-Tool-Bench consists of 1,200 tool-centric problem instances covering data preprocessing, model training, and evaluation stages, each annotated with ground-truth tool APIs and reasoning trajectories. The paper further introduces a lightweight simulator that emulates realistic execution environments and proposes a hierarchical evaluation metric combining plan accuracy, execution success, and tool efficiency. Experiments across a range of models reveal consistent weaknesses in long-horizon tool coordination and parameter grounding.

**Strengths:**

- The paper addresses an underexplored aspect of LLM-for-ML research—tool reasoning—bridging the gap between abstract workflow generation and real tool invocation.
- The dataset is carefully constructed, featuring verified tool APIs and multi-step plans that make evaluation more interpretable and diagnostic than previous black-box setups.

**Weaknesses:**

- While the benchmark is well-motivated, its methodological novelty remains limited—it mainly reorganizes existing task formulations and evaluation paradigms into a cleaner schema, without introducing new modeling or learning components.
- As a benchmark paper, the overall scale feels somewhat limited: both the dataset size and the number of baselines are modest, and the work would be more convincing if it compared tool-based reasoning with direct code-generation approaches (e.g., AutoML or ML-agent frameworks) on the same ML tasks.
- Although the benchmark emphasizes multi-step reasoning, the actual task complexity is relatively constrained—most tasks involve fewer than five tools, with largely deterministic argument structures—restricting generalization to more realistic ML scenarios.
- The paper would also benefit from richer qualitative case studies to illustrate representative reasoning trajectories and better contextualize quantitative results.

**Questions:**

See #Weaknesses

---

> ### Author Response · Authors · 2025-11-24
> **Response to Reviewer f1i6-Part 1**
>
> We sincerely thank the reviewer for their thoughtful feedback and insightful questions. We address their concerns both in the common response section and in the specific responses below.
>
> *1. While the benchmark is well-motivated, its methodological novelty remains limited—it mainly reorganizes existing task formulations and evaluation paradigms into a cleaner schema, without introducing new modeling or learning components.*
>
> We sincerely appreciate the reviewer raising this important point about novelty. We refer the reviewer to the common response on Novelty for a comprehensive discussion. Briefly, ML-Tool-Bench is the first benchmark for long-horizon planning in ML workflows (20-80 sequential tool calls vs. 1-3 in existing tool benchmarks), introducing three methodological innovations necessary for evaluating planning algorithms with complex artifacts.
>
> While we agree that our primary contribution is a benchmark, we respectfully argue that ML-Tool-Bench introduces several methodologically novel elements:
>
> 1. **Named-object management with scratchpad:** Specifically, our scratchpad design (Section 3.1, Figure 1) enables reversible branching in tree search with complex artifacts (dataframes, models). Unlike prior tool benchmarks that restrict arguments to primitive types, our decorator-based system (Section E) allows tools to operate on arbitrarily large objects via named references. This is essential for multi-step ML workflows and enables systematic comparison of search algorithms.
>
> 2. **Stage-based evaluation framework with deterministic verification:** Specifically, our shaped reward stages (Section 4.4) introduce automated verification mechanisms that inspect scratchpad contents and tool messages to deterministically confirm completion criteria (e.g., no NaNs for data cleaning, all columns encoded for feature engineering). This contrasts with subjective LLM-based scoring in LATS [Zhou et al., 2024a], which we show produces inconsistent evaluations as trajectories lengthen, and with heuristic cost functions in Toolchain* [Zhuang et al., 2023]. Our approach provides objective, reproducible rewards while serving dual purposes: guiding search during planning and enabling fine-grained analysis of where agents succeed or fail.
>
> 3. **Hierarchical decomposition with tool masking:** Specifically, our approach (Section 4.5) introduces a systematic methodology for restricting tool visibility per subtask, reducing branching factor while maintaining sequential workflow structure. The enumeration of all solution nodes up to prescribed depth, rather than selecting a single path, avoids locally optimal choices: a design choice distinct from standard hierarchical RL.
>
> We respectfully position our work in the spirit of recent benchmark papers like HumanEval (Chen et al., 2021), MATH (Hendrycks et al., 2021), and MLE-bench (Chan et al., 2025), where the primary contribution is in design of evaluation infrastructure, specifically long trajectory planning in ML-Tool-Bench.
>
> *2. As a benchmark paper, the overall scale feels somewhat limited: both the dataset size and the number of baselines are modest, and the work would be more convincing if it compared tool-based reasoning with direct code-generation approaches (e.g., AutoML or ML-agent frameworks) on the same ML tasks.*
>
> We thank the reviewer for this thoughtful feedback. We address the concerns about dataset size and baseline coverage below.
>
> Regarding dataset size, we refer the reviewer to the common response section on Benchmark Size, which discusses in detail our cost constraints ($5000 for the full evaluation suite) and demonstrates that our 15-challenge scale aligns with similar benchmarks like DS-Agent [Guo et al., 2024] (~20 challenges). Our evaluation prioritizes depth over breadth: 10 independent trajectories per challenge provide robust estimates of success rates for stochastic planning algorithms, where single runs would be unreliable.
>
> Regarding the number of baselines, we respectfully argue that our five planning approaches represent the major paradigms in tool-augmented planning:
> - ReAct [Yao et al., 2023b]: Standard prompting baseline
> - LATS [Zhou et al., 2024a]: Tree search with LLM-based state evaluation
> - MCTS-Outcome: Tree search with outcome-based rewards
> - MCTS-Shaped: Tree search with deterministic shaped rewards
> - Hierarchical MCTS: Hierarchical decomposition with tool masking
>
> These methods systematically explore key design choices: prompting vs. search, LLM-based vs. deterministic evaluation, flat vs. hierarchical decomposition. Our goal is to provide a benchmark for testing planning capabilities in long-horizon tool-augmented settings.

---

> ### Author Response · Authors · 2025-11-24
> **Response to Reviewer f1i6-Part 2**
>
> We sincerely thank the reviewer for their thoughtful feedback and insightful questions. We address their concerns both in the common response section and in the specific responses below. We continue from Part-1.
>
> *3. Although the benchmark emphasizes multi-step reasoning, the actual task complexity is relatively constrained—most tasks involve fewer than five tools, with largely deterministic argument structures—restricting generalization to more realistic ML scenarios.*
>
> We appreciate the reviewer raising this important point about task complexity, and we would like to clarify the benchmark's design.
>
> We respectfully disagree with the characterization that most tasks involve fewer than five tools. Our tool allocation reflects the complexity of real-world machine learning workflows. Critical stages like Feature Engineering involve 30+ tools available to the agent, while Modeling and Data Cleaning each provide approximately 10 tools.
>
> Our successful trajectories typically span 20-30 tool invocations across the full ML pipeline (data loading → cleaning → feature engineering → modeling → prediction). Each subtask in Hierarchical MCTS has access to relevant tools from the full set of 61, and agents must select and sequence multiple tools within each stage. For example, feature engineering alone may require 5-10 tool calls, with each call selecting from 20-30 available feature engineering tools to create derived features, encode categoricals, and normalize numerics before proceeding to modeling.
>
> While tools have defined signatures, the argument identification can be challenging:
> - Hyperparameter selection: Identifying appropriate values for model hyperparameters (e.g., n_estimators, learning_rate, max_depth) from continuous/discrete spaces
> - Column selection: Identifying correct column names from potentially hundreds of features for operations like encoding, aggregation, or feature creation
> - Conditional logic: Many tools require condition specifications (e.g., fillna_with_condition) that depend on understanding data distributions
> - Named object references: Tracking correct intermediate artifacts (dataframes, models) across 20+ tool invocations
>
> This combinatorial complexity—selecting the right tool, specifying correct arguments, and maintaining coherent state across long trajectories—is precisely what makes the planning problem challenging. Our results show that even state-of-the-art methods struggle with this complexity: ReAct [Yao et al., 2023b] and LATS [Zhou et al., 2024a] achieve near-zero performance on many challenges.
>
> We acknowledge the reviewer's point that real-world ML workflows can be even more complex (e.g., custom architectures, ensemble methods, extensive hyperparameter tuning). However, our design deliberately balances realism with tractability:
> - The 61-tool set is sufficient to solve real Kaggle challenges (which we sample from actual competitions)
> - Restricting to tabular tasks and tree-based models enables controlled comparison across methods
> - The tool abstractions reflect common ML libraries (scikit-learn, XGBoost, etc.) used in practice
>
> We appreciate this feedback and view extending to more complex scenarios (larger tool sets, neural architectures) as valuable future work. We would be happy to discuss these limitations more explicitly in the paper.
>
> *4. The paper would also benefit from richer qualitative case studies to illustrate representative reasoning trajectories and better contextualize quantitative results.*
>
> We thank the reviewer for this suggestion. We refer to the common response on Trajectory Analysis and Appendix I, which provide detailed case studies of both successful and failed trajectories.
>
> In Appendix I, we analyze the Spaceship Titanic competition using GPT-4.1-mini across all five planning algorithms. Hierarchical MCTS used 80 tool calls, tested multiple modeling strategies (Random Forest, LightGBM, CatBoost), and reached the 62.65th percentile. In contrast, LATS fit a model but its LLM evaluator preferred suboptimal trajectories, ReAct terminated prematurely, and MCTS-Outcome never reached the modeling phase (0 percentile).
>
> Notably, only Hierarchical MCTS experimented with multiple approaches—all other algorithms created just a single model before evaluation. This demonstrates how different algorithms handle long-horizon reasoning, with Hierarchical MCTS showing systematic exploration while ReAct and LATS exhibit premature convergence and inconsistent value estimation.

---

> > ### Author Response · Authors · 2025-11-26
> > **Response to Reviewer f1i6-Part 3**
> >
> > We sincerely thank the reviewer for their thoughtful feedback and insightful questions. We address their concerns both in the common response section and in the specific responses below. We continue from Part-2.
> >
> > *Comparison with AutoML Benchmarks*
> >
> > We compare our approaches with a direct code-generation variant, ReAct-Code, which produces end-to-end solutions for the challenge and iteratively updates the code based on feedback from execution. The agent is allowed up to 10 iterations to refine its solution and produce a final submission file. We also evaluate all algorithms on the original train and test splits of the Kaggle benchmark, as in Tables 1 and 2, in the paper.
> >
> > We provide the results for Consistency below
> >
> > | Competition                     | ReAct | LATS | MCTS-Outcome | MCTS-Shaped | Hierarchical MCTS | React-Code |
> > |--------------------------------|-------|------|--------------|-------------|-------------------|------------------|
> > | Spaceship Titanic              | 0.3   | 0.0  | 0.6          | 0.6         | 0.4               |1          |
> > | BPM Prediction                 | 0.2   | 0.3  | 0            | 0.8         | 0.7               |1|
> > | Calorie Expenditure Prediction | 0.4   | 0.3  | 0.2          | 0.3         | 0.6               |0.7|
> > | California Housing Regression  | 0.6   | 0.4  | 0.1          | 0.5         | 0.8               |0.8|
> > | Bank Deposit Classification    | 0.4   | 0.5  | 0.0          | 1.0         | 0.6               |0.8|
> > | Bank Churn Classification      | 0.2   | 0.5  | 0.1          | 1.0         | 0.8               |1.0|
> > | **Overall (Median)**               | 0.35  | 0.35 | 0.1          | 0.7         | 0.65              |0.9|
> >
> >
> > We provide the results for the leaderboard percentile below
> >
> > | Competition                     | ReAct | LATS | MCTS-Outcome | MCTS-Shaped | Hierarchical MCTS |React-Code     |
> > |--------------------------------|-------|------|--------------|-------------|-------------------|---------------|
> > | Spaceship Titanic              | 0.0   | 0.0  | 39.77        | 41.69       | 0.0               |48.08   |
> > | BPM Prediction                 | 0     | 0    | 0            | 5.03        | 0.19              |6.75    |
> > | Calorie Expenditure Prediction | 0.0   | 0.0  | 0.0          | 0.0      |           16.81        |24.29    |
> > | California Housing Regression  | 5.51  | 0.0  | 0.0          | 8.99        | 24.78             |29.93 |
> > | Bank Deposit Classification    | 0.0   | 13.47| 0.0          | 27.65       | 27.29             |37.48   |
> > | Bank Churn Classification      | 0.0   | 14.91| 0.0          | 29.87       | 32.47             |35.09    |
> > | **Overall (Median)**             | 0.0   | 0.0  | 0.0          | 18.32       | 20.80             |32.51|
> >
> > ReAct-Code, which generates working code directly to solve the Kaggle challenges, outperforms all planning-oriented methods on both consistency and leaderboard percentile. This result is expected: direct code generation has access to the full modeling and data-processing pipeline, whereas planning methods are constrained to a limited toolset and therefore face a strictly harder problem. Importantly, our planning approaches, despite their restricted tools, do not lag too far behind ReAct-Code’s performance, suggesting that the chosen toolset is largely sufficient for solving tabular Kaggle ML tasks.

---

### Official Review · Reviewer_N6k3 · 2025-10-31

**Soundness:** 2
**Presentation:** 2
**Contribution:** 2
**Rating:** 4
**Confidence:** 3

**Summary:**

The paper presents ML-Tool-Bench, a tool-augmented benchmark for tabular Kaggle-style ML tasks, offering a curated suite of 61 tools and 15 competitions together with an in-memory scratchpad mechanism to persist and reuse intermediate artifacts. The authors evaluate several planning strategies (ReAct, LATS, multiple MCTS variants) and propose two improvements: MCTS-Shaped and Hierarchical MCTS. Experiments on GPT-4o and GPT-4.1-mini show that MCTS-Shaped and Hierarchical MCTS improve trajectory validity and leaderboard percentiles compared to baselines.

**Strengths:**

1. Extending tool-use evaluation to end-to-end ML workflows with long-horizon planning and artifact management.
2. The proposed approach includes multiple complementary features.
3. Results show consistent improvement across different models and metrics.

**Weaknesses:**

1. Manual subtask or tool assignment could bias results. Hierarchical MCTS depends on hand-assigning tools to subtasks, which may imply prior knowledge and limit the generality and automation of the method.

2. The scope of evaluation limited to 15 tabular data challenges. This limits the benchmark's breadth in evaluating a wider range of ML tasks. Furthermore, the benchmark’s 61 tools do not adequately demonstrate scalability to large action spaces.

3. Novelty is insufficient. The method relies on  existing techniques (e.g., task decomposing) without innovations in core mechanisms.

4. The paper emphasizes performance metrics such as success rate and leaderboard percentile but offers little analysis of failure cases or decision dynamics. Without visualizing search trajectories or reward evolution, it is difficult to discern whether performance gains stem primarily from shaped rewards or from hierarchical decomposition.

**Questions:**

1. The description of the Hierarchical MCTS method lacks detail, particularly regarding how it decomposes complex tasks into ordered sequences of subtasks.

2. Computational cost of MCTS and Hierarchical MCTS at larger tool sets or in production settings (latency, budget trade-offs) is not deeply analyzed. How much is the computational overhead and how does it compare to that of the baselines?

3. How does this method apply to non-tabular ML tasks (e.g., Computer Vision, NLP), and what are the scaling strategies and bottlenecks for Hierarchical MCTS when tools far exceed 61?

---

> ### Author Response · Authors · 2025-11-24
> **Response to Reviewer N6k3-Part 1**
>
> We sincerely thank the reviewer for their thoughtful feedback and insightful questions. We address their concerns both in the common response section and in the specific responses below.
>
> *1. Manual subtask or tool assignment could bias results. Hierarchical MCTS depends on hand-assigning tools to subtasks, which may imply prior knowledge and limit the generality and automation of the method.*
>
> We thank the reviewer for raising the question of potential bias from hand assignment for establishing the generality of our approach, and we appreciate the opportunity to clarify it.
>
> 1. **Domain Structure vs. Task-Specific Knowledge**: We want to emphasize that the hierarchical decomposition (Data Loading → Cleaning → Feature Engineering → Modeling → Prediction) is the standard structure across all supervised learning pipelines, not task-specific prior knowledge for our benchmark. This structure is domain-general within ML and does not encode information about specific Kaggle challenges or their solutions. Importantly, the method discovers which tools to use and how to sequence them within each subtask through MCTS exploration, not through manual specification. We acknowledge that this distinction could have been made clearer in our paper and thank the reviewer for highlighting this point.
>
> 2. **Tool Assignment**:
>
>     We want to clarify that tool-to-subtask assignment was performed largely algorithmically:
>       - We provided a state-of-the-art coding agent with subtask descriptions and tool docstrings
>       - The agent performed semantic matching to assign tools to subtasks
>       - One author verified assignments, correcting only minor errors
>
>     We agree with the reviewer that full automation is desirable. This protocol is fully reproducible and domain-agnostic, it requires only natural language descriptions of subtasks and tool docstrings, both of which are available in tool-augmented agentic systems. As LLM capabilities continue to improve, this step will only get easier.
>
> *2. The scope of evaluation limited to 15 tabular data challenges. This limits the benchmark's breadth in evaluating a wider range of ML tasks. Furthermore, the benchmark's 61 tools do not adequately demonstrate scalability to large action spaces.*
>
> We are grateful to the reviewer for this insightful comment, which touches on two important aspects of our evaluation: the number of challenges and the action space size. We address each point below.
>
> For the question regarding benchmark coverage, we refer the reviewer to our common response. In summary, our evaluation required substantial investment ($5000 USD for the full suite) and aligns with recent benchmark papers of similar scale (DS-Agent [Guo et al., 2024]).
>
> We argue that 61 tools represents a significantly large action space for tool-augmented LLM planning, particularly when considering the combinatorial complexity involved:
>
>   - Tool selection complexity: The agent must select relevant tools at each stage (out of 61 total).
>   - Argument specification complexity: Each tool requires correct parameterization from continuous and discrete spaces (e.g., hyperparameters, column names, preprocessing options)
>   - Compositional complexity: The agent must discover effective tool sequences, leading to exponential growth in the search space
>
> Further, there do not exist any planning benchmarks for LLMs currently that involve close to 61 tools. Existing benchmarks (see BFCL [Patil et al., 2025], ToolBench [Xu et al., 2023]) mainly focus on identifying the correct tool to call, alongside correct tool arguments.
>
> *3. Novelty is insufficient. The method relies on existing techniques (e.g., task decomposing) without innovations in core mechanisms.*
>
> We appreciate the reviewer raising this important concern about novelty. We refer the reviewer to the common response on Novelty for comprehensive details. Briefly, our primary contribution is the first benchmark specifically designed to evaluate long-horizon planning capability in tool-augmented LLMs for ML workflows, requiring 20-80 sequential tool calls compared to 1-3 in existing tool benchmarks. To support this evaluation, we introduce three methodological innovations: scratchpad-augmented planning with named-object management, deterministic shaped rewards with stage-based verification, and hierarchical decomposition with tool masking—each addressing unique challenges of evaluating planning algorithms with complex artifacts.

---

> > ### Author Response · Authors · 2025-11-24
> > **Response to Reviewer N6k3-Part 2**
> >
> > We sincerely thank the reviewer for their thoughtful feedback and insightful questions. We address their concerns both in the common response section and in the specific responses below. We continue from Part-1.
> >
> > *4. The paper emphasizes performance metrics such as success rate and leaderboard percentile but offers little analysis of failure cases or decision dynamics. Without visualizing search trajectories or reward evolution, it is difficult to discern whether performance gains stem primarily from shaped rewards or from hierarchical decomposition.*
> >
> > We greatly appreciate this feedback and agree that trajectory analysis strengthens our evaluation. We refer the reviewer to the common response on Trajectory Analysis and Appendix I for detailed case studies. In summary, we provide comprehensive analysis of both successful and failed trajectories across all planning variants for the Spaceship Titanic competition using GPT-4.1-mini. The case study reveals that Hierarchical MCTS explored 80 tool calls and tested multiple modeling strategies (Random Forest, LightGBM, CatBoost), while other algorithms created only a single model before evaluation—demonstrating how hierarchical decomposition enables systematic exploration versus premature convergence in flat approaches.
> >
> > *5. The description of the Hierarchical MCTS method lacks detail, particularly regarding how it decomposes complex tasks into ordered sequences of subtasks.*
> >
> > We thank the reviewer for this valuable feedback. The Hierarchical MCTS method is described in Section 4.5 and illustrated in Figure 3. However, we acknowledge the reviewer's point that additional implementation details would strengthen the paper. We provide additional details for clarity:
> >
> > Hierarchical MCTS decomposes ML challenges into ordered subtasks that reflect the standard supervised learning pipeline: Data Loading → Cleaning → Feature Engineering → Modeling → Prediction. The decomposition is similar to the reward shaping structure, described in the section on MCTS-Shaped in the paper.
> >
> > Our approach proceeds sequentially through these subtasks. At each subtask, the agent performs MCTS exploration restricted to the subset of tools relevant to that stage (20-30 tools per subtask from the full 61-tool set). Once the subtask's completion criteria are met (e.g., no missing values remain after Data Cleaning), the agent transitions to the next subtask with a new MCTS tree. This decomposition reduces the branching factor at each decision point while maintaining the sequential structure required by ML workflows, enabling more efficient exploration compared to a flat MCTS over the entire 61-tool space.
> >
> > *6. Computational cost of MCTS and Hierarchical MCTS at larger tool sets or in production settings (latency, budget trade-offs) is not deeply analyzed. How much is the computational overhead and how does it compare to that of the baselines?*
> >
> > We thank the reviewer for this important question about computational efficiency. We refer the reviewer to the common response on Benchmark Size for detailed cost analysis across all methods. In brief, our evaluation suite cost $5000 total for the full experimental setup, with LATS being 3.5x more expensive than our successful variants (Hierarchical-MCTS and MCTS-Shaped) while achieving only 0.2 consistency and 0 median percentile across five Kaggle challenges. This demonstrates that Hierarchical MCTS's performance gains strongly justify its computational cost: it outperforms both expensive but ineffective methods (LATS) and cheap but poorly performing ones (ReAct, which costs 10.5x less than LATS but achieves similar near-zero performance).
> >
> > *7. How does this method apply to non-tabular ML tasks (e.g., Computer Vision, NLP), and what are the scaling strategies and bottlenecks for Hierarchical MCTS when tools far exceed 61?*
> >
> > We thank the reviewer for this important question about the broader applicability and scalability of our approach. While our current evaluation focuses on tabular ML for tractability, the planning challenges we evaluate—tool coordination, artifact management, multi-step reasoning—are domain-agnostic and apply to any multi-step ML workflow. Extending to non-tabular modalities (computer vision, NLP) would require substantial changes to the tool design, to handle: different modalities, model architectures, and training pipelines.
> >
> > It has been observed that as we increase the number of tools available to the agent, the performance significantly deteriorates [1]. This empirical finding motivates our hierarchical decomposition approach, which bounds the effective action space at each stage. Our ablation in Appendix F demonstrates that tool masking is critical for scaling (median consistency drops from 0.8 to 0.3 without it), providing strong evidence that hierarchical approaches enable scaling to larger toolsets.
> >
> > [1] Using Invocable APIs derived from NL2SQL datasets for LLM Tool-Calling Evaluation, Elder et al., 2025.

---

### Official Review · Reviewer_SNrZ · 2025-11-01

**Soundness:** 2
**Presentation:** 3
**Contribution:** 3
**Rating:** 4
**Confidence:** 4

**Summary:**

The paper presents ML-Tool-Bench, a tool-augmented benchmark for end-to-end tabular ML workflows drawn from 15 Kaggle challenges. Agents operate with a named-object scratchpad and a curated set of 61 tools that cover loading, cleaning, feature engineering, modeling, and prediction. The study compares multiple planning frameworks, including ReAct, LATS, MCTS variants with shaped rewards, and a hierarchical MCTS that performs subtask decomposition with tool masking. Results are reported as leaderboard percentiles and “consistency” of valid trajectories per task and model.

**Strengths:**

1. The task is very relevant. Real ML engineering requires multi-step planning, reasoning, and execution, and the benchmark targets this long-horizon setting with artifact reuse. The scratchpad design directly addresses state corruption issues in multi-step pipelines.

2. The setup of representing ML steps as tools reduces emphasis on raw coding and focuses evaluation on the structure of the ML pipeline. The toolset is clearly scoped across the main stages of tabular ML.

3. The experiments cover several agent frameworks and explore targeted reward signals. Modeling the pipeline with decomposition and shaped rewards is shown to improve performance on many tasks.

**Weaknesses:**

1. Benchmark size - With only 15 tasks the variance is high across problems, which makes aggregate comparisons unstable. The tables show very large swings in percentile across tasks, including cases with near 0 and near 100 for the same methods on different tasks. I would suggest expanding to atleast 30–50 tasks per task family to produce a more reliable signal, and claims should be scoped to tabular ML for now.

2. Data interaction clarity - Many tools target code-level operations like loading dataframes, encoding, and fitting models. For the models to truly reason about the right pipeline or algorithm for the task, it would be important to review the data or run additional analysis. It is unclear if the model can do that and if so, how is it instantiated especially considering model context lengths.

3. Comparability to public leaderboards - Training is capped at 10k examples per task for compute constraints, but this makes the results not comparable with the public leaderboard. The paper should add cross-validation and a small public-LB calibration to validate if this comparison holds.

4. Ablations - Hierarchical MCTS bundles subtasking, tool masking, and shaped rewards. Isolating each component with a small factorial ablation would clarify where the gains come from.

**Questions:**

Please include the exact prompts for each algorithmic setup and the tool-calling templates in the appendix to improve reproducibility.

---

> ### Author Response · Authors · 2025-11-24
> **Response to Reviewer SNrZ-Part 1**
>
> We sincerely thank the reviewer for their thoughtful feedback and insightful questions. We address their concerns both in the common response section and in the specific responses below.
>
> *1. Benchmark size. With only 15 tasks the variance is high across problems, which makes aggregate comparisons unstable. The tables show very large swings in percentile across tasks, including cases with near 0 and near 100 for the same methods on different tasks. I would suggest expanding to atleast 30–50 tasks per task family to produce a more reliable signal, and claims should be scoped to tabular ML for now.*
>
> We appreciate the reviewer's concern about variance and its impact on aggregate comparisons. We want to clarify that the variance reflects genuine difficulty differences across Kaggle challenges: some require 20 steps while others require 30+, with varying tool requirements and solution complexity. To address the reliability of our evaluation under this variance, we refer the reviewer to Section 5.2, which introduces our consistency metric, the percentage of valid successful trajectories, designed specifically to measure reliability across stochastic planning algorithms. Despite variance in individual challenge difficulty, our results show consistent performance gaps across methods: Hierarchical MCTS outperforms baselines on 13/15 challenges (87%) and MCTS-Shaped on 11/15 (73%), while ReAct and LATS struggle consistently across most challenges. This demonstrates that while individual challenges vary in difficulty, the relative performance rankings remain stable and reliable. We thank the reviewer for the suggestion to scope our claims to tabular ML, and we agree with this recommendation.
>
> We refer the reviewer to our common response on Benchmark Size for detailed discussion. In summary, we restricted the benchmark to 15 challenges due to cost constraints ($5000 for 10 trajectories × 5 approaches × 15 challenges × 2 models), which aligns with similar benchmarks like DS-Agent [Guo et al., 2024] that used ~20 challenges. Our 15-challenge scale also aligns with other recent benchmarks involving expensive evaluation such as MLAgent-bench [Huang et al., 2024] (13 challenges) and MLE-bench [Chan et al., 2025] (75 challenges, but single-pass code generation not iterative planning with tree search). Benchmarks with larger scales (e.g., HumanEval: 164 problems) evaluate inexpensive operations (single code generation pass) rather than multi-step planning requiring hundreds of LLM calls per trajectory. Our evaluation prioritizes depth over breadth: 10 independent trajectories per challenge provide robust estimates of each method's success rate, measured using consistency metric (Section 5.2), which is crucial for stochastic planning algorithms where single runs are unreliable.
>
> *2. Data interaction clarity. Many tools target code-level operations like loading dataframes, encoding, and fitting models. For the models to truly reason about the right pipeline or algorithm for the task, it would be important to review the data or run additional analysis. It is unclear if the model can do that and if so, how is it instantiated especially considering model context lengths.*
>
> We thank the reviewer for this insightful question about how our agents reason about data properties. Our tool library includes several data inspection capabilities that enable the agent to analyze data characteristics before making pipeline decisions. These tools provide summary statistics (get_summary_statistics), examine data distributions, and inspect missing value patterns (get_missing_values_info). The outputs from these tools are returned as structured text messages that fit within context windows, rather than displaying raw dataframes. This design allows the agent to make informed decisions about preprocessing strategies, feature engineering approaches, and model selection based on concrete data properties.

---

> ### Author Response · Authors · 2025-11-24
> **Response to Reviewer SNrZ-Part 2**
>
> We sincerely thank the reviewer for their thoughtful feedback and insightful questions. We address their concerns both in the common response section and in the specific responses below. We continue from Part-1.
>
> *3. Comparability to Public Leaderboard*
>
> We use the original train and test splits for each Kaggle challenge across a subset of six tasks from our benchmark, and evaluate all planning algorithms using GPT-4.1-mini as the backbone. The results for these experiments are provided in Tables 1 and 2. We observe similar performance trends to those seen on our benchmark, which relies on subsampled data: the relative behavior of the planning algorithms remains consistent when evaluated using both the Kaggle public leaderboard and the full original datasets. This suggests that performance on our benchmark serves as a reasonable proxy for estimating performance on the actual Kaggle public leaderboard when using the unfiltered train and test data.
>
> *4. Ablations. Hierarchical MCTS bundles subtasking, tool masking, and shaped rewards. Isolating each component with a small factorial ablation would clarify where the gains come from.*
>
> We want to clarify that our approach *Hierarchical MCTS* does not use shaped rewards. In our work, only *MCTS-Shaped* approach uses shaped rewards. We will add additional details in sections 4 & 5 to make it very clear.
>
> We appreciate the reviewer's suggestion for conducting factorial ablations to isolate individual components. To address this concern, we conducted an ablation study that removes tool masking from Hierarchical MCTS while preserving the hierarchical subtask decomposition. In this ablation, the agent still proceeds through ordered subtasks (Data Loading → Cleaning → Feature Engineering → Modeling → Prediction) but has access to all 61 tools at every stage, rather than the subset of 20-30 tools relevant to each subtask.
>
> The results in Appendix F demonstrate that tool masking is critical for performance. Specifically, Hierarchical MCTS without tool masking achieves a median consistency of 0.3 and a median leaderboard percentile position of 0 across a subset of five challenges. In comparison, Hierarchical MCTS with tool masking achieves a median consistency and leaderboard percentile of 0.8 and 21.10 respectively. This substantial performance degradation highlights that both components—hierarchical subtask decomposition and tool masking—are critical for effectively solving long-horizon planning problems in high-dimensional action spaces using LLMs. The performance improvement comes from the combination of these components: hierarchical structure provides workflow guidance, while tool masking reduces the branching factor at each decision point, enabling more focused exploration.
>
> *5. Please include the exact prompts for each algorithmic setup and the tool-calling templates in the appendix to improve reproducibility.*
>
> We thank the reviewer for the suggestion and we have added all the prompts used in our approaches in Appendix.

---

### Author Response · Authors · 2025-11-24
**Common Reviewer Response-Novelty**

We thank all the reviewers for their detailed and thoughtful feedback on our submission and appreciate the opportunity to clarify their questions. Here we address the common question raised by the reviewers, regarding the novelty of our benchmark.

## Novelty

We want to highlight that the primary contribution is the **first**  benchmark specifically designed to evaluate the long trajectory planning capability of LLMs in complex, multi-step ML workflows. This represents a significant departure from existing benchmarks:

  - Most ML benchmarks (MLAgent-bench [Huang et al., 2024], MLE-bench [Chan et al., 2025], MLE-dojo [Qiang et al., 2025], DS-Agent [Guo et al., 2024], AIDE [Jiang et al., 2025]) all focus on direct code generation
  - Existing tool benchmarks (ToolBench [Xu et al., 2023], BFCL [Patil et al., 2025], τ-Bench [Yao et al., 2024]) focus on tool selection and argument extraction

Our benchmark requires multi-step planning, tool composition, and sequential decision-making across 20+ step solution trajectories, compared to 1-3 steps in existing tool benchmarks, reaching a maximum of around 80 steps, without explicit guidance. This distinction is crucial: we evaluate whether LLMs can plan complex solutions, not merely execute predefined steps. To our knowledge, no existing benchmark provides this capability for tool-augmented LLMs in realistic ML workflows. To make this distinction concrete, we provide a comparison:

| Benchmark | Focus | Tool Calls | Planning Depth | Artifact Management |
|-----------|-------|------------|----------------|---------------------|
| ML-Tool-Bench (Ours) | Multi-step planning | 20-30 | Deep | Complex objects |
| ToolBench (Xu et al., 2023) | Tool selection | 1-3 | Shallow | String-based |
| BFCL (Patil et al., 2025) | Argument extraction | 1-3 | Shallow | Primitive types |
| MLE-bench (Chan et al., 2025) | Code generation | N/A | N/A | File system |
| MLAgent-bench (Huang et al., 2024) | Code execution | N/A | N/A | File system |

As summarized in the table above, ML-Tool-Bench's unique focus on long-horizon planning (20+ tool calls) with complex artifact management distinguishes it from existing benchmarks. To validate this benchmark and establish baseline performance, we provide a comprehensive analysis of existing planning methods (ReAct [Yao et al., 2023b], LATS [Zhou et al., 2024a], reward shaping, hierarchical decomposition). Our evaluation in Section 5 demonstrates that:

  - State-of-the-art methods (ReAct [Yao et al., 2023b], LATS [Zhou et al., 2024a]) struggle on deeper planning problems
  - Two simple techniques (reward shaping, hierarchical MCTS) significantly outperform baselines

We view our work as primarily a **benchmark contribution** that enables future algorithmic research, similar in spirit to influential benchmarks like HumanEval [Chen et al., 2021] for code generation or MATH [Hendrycks et al., 2021] for mathematical reasoning. The algorithmic comparisons serve to validate the benchmark's utility and establish baseline performance for future work.

To support this benchmark evaluation, we introduce three methodological innovations that address the unique challenges of long-horizon tool planning:

1. **Scratchpad-augmented planning with named-object management** (Section 3.1): Enables reversible branching in tree search with complex artifacts (dataframes, models). Unlike prior work that restricts tool arguments to primitive types, our approach handles arbitrarily large objects.

2. **Deterministic shaped rewards with stage-based verification** (Section 4.4): Replaces subjective LLM-based scoring with automated verification that inspects scratchpad contents. This provides objective, reproducible rewards crucial for search algorithms.

3. **Hierarchical decomposition with tool masking** (Section 4.5): Systematically reduces branching factor while maintaining workflow structure, enumerating all solution nodes till a maximum depth, rather than selecting single paths.

These contributions constitute essential infrastructure for evaluating planning algorithms in tool-augmented settings and distinguish our work from pure benchmarking efforts. We hope this clarification addresses the reviewers' concerns regarding novelty. We would be happy to restructure the paper to more clearly emphasize both the benchmark contributions and the methodological innovations.

---

> ### Author Response · Authors · 2025-11-24
> **Common Reviewer Response - Benchmark Size and Trajectory Analysis**
>
> We thank all the reviewers for their detailed and thoughtful feedback on our submission and appreciate the opportunity to clarify their questions. Here, we address the common question raised by the reviewers regarding benchmark size and trajectory analysis.
>
> ## Benchmark Size
>
> We agree with the reviewers that having more challenges would make the benchmark more robust. However, we want to highlight that we restricted it to 15 challenges for cost reasons. For our evaluation suite, we generate 10 trajectories for each of the 5 approaches, for 15 challenges, and 2 models. The total cost was 5000 USD. We note that the scale of the benchmark is similar to previous benchmarks such as DS-Agent (see [1]), where around 20 challenges were used.
>
> ### Cost Comparison on Five Kaggle Challenges
>
> We also provide cost comparisons for all planning algorithms using GPT-4.1-mini on the same subset of five Kaggle challenges from our benchmark. The results are reported in Table below. LATS is the most expensive planning approach, costing 3.5x more than the more successful variants (Hierarchical-MCTS and MCTS-Shaped), while achieving only a consistency of 0.2 and a median percentile position of 0 across the five Kaggle challenges. This suggests that LATS's search is unfocused due to inconsistent scoring by the LLM evaluator. ReAct is the cheapest method but also performs poorly, only marginally outperforming LATS despite the latter costing 10.5x more.
>
>
>
> Total costs (USD $) aggregated over 10 trajectories for each of the five competitions using GPT-4.1-mini under all planning algorithm variants examined in this study. For each row, the maximum value is highlighted in bold.
>
> | Competition | ReAct | LATS-Reflection | MCTS-Outcome | MCTS-Shaped | Hierarchical MCTS |
> |-------------|-------|-----------------|--------------|-------------|-------------------|
> | Spaceship Titanic | 1.66 | **22.29** | 3.22 | 8.44 | 3.93 |
> | Poisonous Mushroom Prediction | 0.86 | **13.64** | 3.13 | 2.86 | 1.89 |
> | Bank Churn Classification | 1.21 | **9.98** | 1.44 | 1.79 | 5.65 |
> | Santander Customer Transaction Prediction | 1.34 | **11.28** | 2.36 | 1.43 | 5.38 |
> | NYC Taxi Fare Prediction | 2.02 | **17.83** | 2.72 | 4.07 | 5.56 |
> | **Sum Total Cost** | **7.08** | **75.02** | **12.86** | **18.59** | **22.42** |
>
> [1] DS-Agent: Automated Data Science by Empowering Large Language Models with Case-Based Reasoning, Guo et al., ICML 2024.
>
>
> ## Trajectory Analysis
>
> We refer the reviewers to Appendix I for example failed and successful trajectories across all planning variants. We also briefly summarize the key findings from Appendix I below.
>
> ### Case study in Appendix I
>
> The case study in Appendix I analyzes the Spaceship Titanic Kaggle competition using GPT-4.1-mini across all five planning algorithms.
>
>
> | Algorithm | Failed Trajectory Tool Calls | Successful Trajectory Tool Calls | Successful Trajectory Performance |
> |-------------|-------|-----------------|--------------|
> | ReAct| 12 | 40 | 54.92 |
> | LATS | 17 | 27 | 58.21 |
> | MCTS-Outcome | 18 | 42 | 0 |
> | MCTS-Shaped | 19 | 23 | 33.11 |
> | Hierarchical-MCTS | 55 | 80 | 62.65 |
>
> In the table above, the tool calls column denotes the number of tool calls made by the agent in the respective trajectory. The successful trajectory performance denotes the leaderboard percentile position of the submission file created by the successful trajectory corresponding to the planning algorithm.
>
> Across all algorithms except Hierarchical MCTS, only a single valid model was created before the agent moved on to the evaluation phase. In Hierarchical MCTS, the agent experimented with multiple machine learning models (Random Forest, LightGBM, CatBoost, etc.) and selected the best one. We note that the failed trajectory in LATS actually fits a model, but the LLM evaluator preferred other suboptimal trajectories, pointing to issues with LLM state estimation. Failed trajectories in ReAct also displayed premature termination before tasks were completed. We observed that some failed trajectories were unable to call tools successfully or struggled to reach the modeling phase due to lack of guidance (MCTS-Outcome). We point out that the successful trajectory for MCTS-Outcome in our case study created a submission file, but it has a shape mismatch with the original test data and was consequently rated with a percentile position of 0. We highlight that the example successful trajectory of Hierarchical MCTS used 80 tool calls, explored multiple modeling strategies, and achieved a leaderboard percentile position of 62.65.

---

### Author Response · Authors · 2025-12-04
**Response to Area Chairs**

We sincerely thank you and all reviewers for the significant time and effort dedicated to reviewing our work. To facilitate the meta-review process, we briefly summarize the novelty and contributions of our paper, reviewer discussions, and the paper revisions made during rebuttal.

## Novelty & Contributions

Our main contribution is introducing the first benchmark that evaluates long-horizon, tool-augmented planning capabilities of LLMs in complex, multi-step ML workflows involving arbitrarily complex artifacts. Existing ML benchmarks (MLE-Bench, MLAgent-Bench, MLE-Dojo, DS-Agent) primarily assess direct code generation, while current tool-use benchmarks (BFCL, ToolBench, τ-Bench) focus on tool selection or argument extraction. Furthermore, ML-Tool-Bench supports rich data types, such as models and dataframes, whereas prior benchmarks limit tools to primitive arguments.

Our benchmark requires multi-step planning, tool composition, and sequential decision-making over 20+ step trajectories, with some observed cases reaching ~80 steps. ML-Tool-Bench assess whether LLMs can plan complex workflows, rather than simply execute predefined actions.

We further demonstrate that existing planning methods like ReAct and LATS struggle with long-horizon problems. We introduce two simple strategies, reward shaping and task decomposition, and show that they substantially outperform LATS, which suffers from inconsistent LLM-based scoring, and ReAct, which often becomes stuck and fails to self-correct, in terms of median leaderboard percentile. Our proposed approaches also achieve significantly better performance at **3.5× lower cost** compared to LATS. When evaluating costs across a subset of five Kaggle challenges from our benchmark using **GPT-4.1-mini**, LATS incurred a total cost of **\$75** while obtaining a **median leaderboard percentile of 0**. In contrast, our approaches cost approximately **\$20** and achieved a **median leaderboard percentile of around 20**.


## Reviewer Discussions

All reviewers agreed that the benchmark is relevant and that our proposed approaches show clear improvements over existing baselines across multiple models and metrics. Reviewers **tCJh** and **N6k3** specifically highlighted the value of our scratchpad-based object management system, which enables the model to reuse intermediate results and recover from corrupted data. Reviewer **f1i6** appreciated the clarity of our tool API specification, noting that it offers interpretability absent in black-box benchmark designs.

We addressed concerns about the benchmark’s novelty and size in both the common response and our individual replies. We clarified that our work introduces the **first benchmark designed to evaluate long-horizon planning for tool-augmented LLMs in ML workflows**. Regarding the benchmark scale, we argued that the current size is sufficient to reveal consistent performance differences between planning methods, aligns with cost constraints, and is comparable to the number of tasks used in related benchmarks.

In response to reviewer feedback, we added results using the original train/test splits for each challenge and included performance on Kaggle public leaderboards (Tables 1 and 2), which show the same trends observed on our benchmark. Addressing reviewer questions, we also added:

- **Tool-masking ablations** (Appendix F)
- **Cost comparisons** (Appendix G)
- **Trajectory visualizations** illustrating how different planning approaches search for solutions (Appendix I)
- Additional details on **Hierarchical MCTS**, including how tools are assigned to subtasks

These additions directly address the reviewers’ questions and further clarify the benchmark’s design and contributions.


We hope the Area Chairs will take into account that we provided substantial results and explanations addressing the weaknesses and questions raised by the reviewers, especially given that we did not receive responses from most reviewers.


## Revision Summary
- Additional details explaining Hierarchical MCTS on Page 9
- Experiments showcasing the performance of tool-trajectories on Kaggle public leaderboard, when using the original unfiltered train and test data from Kaggle, on Page 10
- Tool Masking ablations in Appendix F
- Cost comparision between approaches in Appendix G
- Prompts used by the planning approaches in Appendix H
- Trajectory Case Study, wherein we visualize tool trajectories proposed by different algorithms, in Appendix I

---

### Meta-Review · Area_Chair_zkft · 2026-01-04

**Summary:**

Reviewers broadly agree that the paper targets a relevant and underexplored problem: evaluating long-horizon planning for tool-augmented ML agents. However, there is a strong and consistent concern that the contribution is primarily incremental. While the benchmark setting is useful, its scale is limited, and the methodological contributions are viewed as engineering refinements of existing ideas (task decomposition, reward shaping, MCTS variants) rather than conceptually new techniques. Multiple reviewers question whether the benchmark and algorithms rise to the level of novelty expected for ICLR.

**Reviewer Concerns:**

1. Common strengths acknowledged by all reviewers:

   * The benchmark addresses long-horizon, multi-step ML workflows, which are not well covered by existing tool-use benchmarks.
   * The scratchpad-based named object management is viewed as a practical and helpful design choice.
   * Empirical results consistently show improvements over ReAct and LATS under the proposed settings.

2. Major concerns that were partially but not fully resolved:

   * Novelty: Reviewers N6k3, f1i6, and tCJh explicitly state that the work largely recombines known ideas (hierarchical decomposition, MCTS, reward shaping) without introducing new core mechanisms. The rebuttal clarified positioning but did not fundamentally change this perception.
   * Benchmark scale and scope: Several reviewers (SNrZ, N6k3, f1i6) remain unconvinced that 15 tabular tasks are sufficient to support strong general claims, despite cost-based justification.
   * Generality and bias: Concerns remain about manual or semi-manual subtask and tool assignment in Hierarchical MCTS, and about how well the benchmark extrapolates beyond tabular ML.
   * Conceptual clarity: Reviewer tCJh explicitly noted that the paper’s core purpose and distinction from existing benchmarks were not sufficiently clear in the main text, and maintained their score after rebuttal.

3. Concerns largely addressed in rebuttal:

   * Reproducibility: Prompts, cost breakdowns, tool masking ablations, and trajectory analyses were added and generally acknowledged as improvements.
   * Cost analysis: The rebuttal clarified that the proposed methods are cheaper and more stable than LATS, addressing efficiency questions.

Overall, while the rebuttal strengthened soundness and presentation, it did not materially change reviewer opinions on novelty and scope.

**Reviewer Scores:**

* Reviewer SNrZ: Likely unchanged. Initially marginally below threshold with openness to acceptance, but core concerns about scale and ablations remain.
* Reviewer N6k3: Unlikely to increase. Explicitly maintains that novelty is insufficient and scope is limited.
* Reviewer f1i6: Unlikely to increase. Views the work as a well-organized benchmark with limited methodological novelty.
* Reviewer tCJh: Explicitly stated they would maintain their score despite rebuttal.

---

### Decision · Program_Chairs · 2026-01-26

Reject